# Conserved chromatin and repetitive patterns reveal slow genome evolution in frogs

Jessen V. Bredeson [1,2,19], Austin B. Mudd[1,19], Sofia Medina-Ruiz[1,19], Therese Mitros[1], Owen Kabnick Smith [3], Kelly E. Miller[1], Jessica B. Lyons [1], Sanjit S. Batra[4], Joseph Park[1], Kodiak C. Berkoff [1], Christopher Plott [5], Jane Grimwood [5], Jeremy Schmutz [5], Guadalupe Aguirre-Figueroa[3], Mustafa K. Khokha [6], Maura Lane[6], Isabelle Philipp[1], Mara Laslo [7], James Hanken [7], Gwenneg Kerdivel[8], Nicolas Buisine[8], Laurent M. Sachs [8], Daniel R. Buchholz[9], Taejoon Kwon [10,11], Heidi Smith-Parker [12], Marcos Gridi-Papp[13], Michael J. Ryan[12], Robert D. Denton[14], John H. Malone [14], John B. Wallingford[15], Aaron F. Straight [3], Rebecca Heald [1], Dirk Hockemeyer [1,16,17], Richard M. Harland [1] & Daniel S. Rokhsar [1,2,16,17,18] ✉

Frogs are an ecologically diverse and phylogenetically ancient group of anuran amphibians that include important vertebrate cell and developmental model systems, notably the genus *Xenopus*. Here we report a high-quality reference genome sequence for the western clawed frog, *Xenopus tropicalis*, along with draft chromosome-scale sequences of three distantly related emerging model frog species, *Eleutherodactylus coqui*, *Engystomops pustulosus*, and *Hymenochirus boettgeri*. Frog chromosomes have remained remarkably stable since the Mesozoic Era, with limited Robertsonian (i.e., arm-preserving) translocations and end-to-end fusions found among the smaller chromosomes. Conservation of synteny includes conservation of centromere locations, marked by centromeric tandem repeats associated with Cenp-a binding surrounded by pericentromeric LINE/L1 elements. This work explores the structure of chromosomes across frogs, using a dense meiotic linkage map for *X. tropicalis* and chromatin conformation capture (Hi-C) data for all species. Abundant satellite repeats occupy the unusually long (~20 megabase) terminal regions of each chromosome that coincide with high rates of recombination. Both embryonic and differentiated cells show reproducible associations of centromeric chromatin and of telomeres, reflecting a Rabl-like configuration. Our comparative analyses reveal 13 conserved ancestral anuran chromosomes from which contemporary frog genomes were constructed.

Amphibians are widely used models in developmental and cell biology[1–5], and their importance extends to the fields of infectious disease, ecology, pharmacology, environmental health, and biological diversity[6–10]. While the principal model systems belong to the genus *Xenopus* (notably the diploid western clawed frog *X. tropicalis* and the paleo-allotetraploid African clawed frog *X. laevis*), other amphibian models have increasingly been introduced due to their diverse developmental, cell biological, physiological, and behavioral adaptations[11–21].

While genome evolution has been extensively studied in mammals[22] and birds[23,24], the relative lack of phylogenetically diverse chromosome-scale frog genomes has limited the study of genome evolution in anuran amphibians. Here, we report a high-quality assembly for *X. tropicalis* and three new chromosome-scale genome assemblies for the Puerto Rican coquí (*Eleutherodactylus coqui*), a direct-developing frog without a tadpole stage[16,19], the túngara frog (*Engystomops pustulosus*), which is a model for vocalization and mate choice[15,18,20], and the Zaire dwarf clawed frog (*Hymenochirus boettgeri*), which has an unusually small embryo, is a model for regulation of cell and body sizes, and a source of potent host-defense peptides with therapeutic potential[13,17,21]. Genome assemblies are essential resources for further work to exploit the experimental possibilities of these diverse animals. The new high-quality *X. tropicalis* genome upgrades previous draft assemblies[25,26] and our new genomes complement draft chromosome-scale sequences for the African clawed frog[27] (*Xenopus laevis*), the African bullfrog[28] (*Pyxicephalus adspersus*), the Leishan moustache toad[29] (*Leptobrachium leishanense*), the Ailao moustache toad[30] (*Leptobrachium* [*Vibrissaphora*] *ailaonicum*), and Asiatic toad[31] (*Bufo gargarizans*), as well as scaffold- and contig-scale assemblies for other species[32]. The rapidly increasing number of chromosome-scale genome assemblies makes anurans ripe for comparative genomic and evolutionary analysis.

Chromosome number variation among frogs is limited[33–35]. Based on cytological[36,37] and sequence comparisons[25,27,33,38,39] most frogs have *n* ~10–12 pairs of chromosomes. A recent meiotic map of the yellow-bellied toad *Bombina variegata* showed that its twelve chromosomes are simply related to the ten chromosomes of *X. tropicalis*[40]. The stability of the frog karyotype contrasts with the more dramatic variation seen across mammals[22,37,41,42], which as a group is considerably younger than frogs. The constancy of the frog karyotype parallels the static karyotypes of birds[23,43], although birds typically have nearly three times more chromosomes than frogs, including numerous microchromosomes (among frogs, only the basal *Ascaphus*[44] has microchromosomes). Despite the stable frog chromosome number, however, fusions, fissions, and other interchromosomal rearrangements do occur, and we can use comparisons among chromosome-scale genome sequences to (1) infer the ancestral chromosomal elements, (2) determine the rearrangements that have occurred during frog phylogeny, and (3) characterize the patterns of chromosomal change among frogs. These findings of conserved synteny among frogs are consistent with prior demonstrations of conservation between *Xenopus tropicalis* with other tetrapods, including human and chicken[25,45].

Since frog karyotypes are so highly conserved, *X. tropicalis* can be used as a model for studying chromosome structure[40], chromatin interaction, and recombination for the entire clade. Features that can be illuminated at the sequence level include the structure and organization of centromeres and the nature of the unusually long subtelomeres relative to mammals (frog subtelomeres are ~20 megabases, compared with the mammalian subtelomeres that are typically shorter than a megabase). The extended subtelomeres of frogs form interacting chromatin structures in interphase nuclei that reflect three-dimensional intra-chromosome and inter-chromosome subtelomeric contacts, which are consistent with a "Rabl-like" configuration. As in other animals, subtelomeres of frogs have an elevated GC content and recombination rate. Here we show that the unusually high enrichment of recombination in the subtelomeres likely reflects similar structural and functional properties in other vertebrates, though the quality of the assembly reveals that the length of subtelomeres, expansion of microsatellite repeat sequences by unequal crossing over, and high recombination rates are considerably greater in frogs than in mammals. A strong correlation between recombination rate and microsatellite sequences suggests that unequal crossing over during meiotic recombination is implicated in the expansion of satellites in the subtelomeres. We use Cenp-a binding at satellites to confirm centromere identity and extend the predictive power of the repeat structures to centromeres of other frogs. We address the unusually high recombination rate in subtelomeric regions, correlating with the landscape of base composition and transposons. Over the 200 million years (My) of evolution that we address here, centromeres have generally been stable, but the few karyotypic changes reveal the predominant Robertsonian translocations at centromeric regions; we also document the slow degeneration that occurs to inactivated centromeres and fused telomeres, changes that are obscured in animals with rapidly evolving karyotypes.

## Results and discussion
### High-quality chromosome-scale genome assembly for *X. tropicalis*

To study the structure and organization of *Xenopus tropicalis* chromosomes and facilitate comparisons with other frog genomes, we assembled a high-quality chromosomal reference genome sequence (Supplementary Data 1, Supplementary Fig. 1, and Supplementary Notes 1 and 2) by integrating data from multiple sequencing technologies, including Single-Molecule Real-Time long reads (SMRT sequencing; Pacific Biosciences), linked-read sets (10x Genomics), short-read shotgun sequencing, in vivo chromatin conformation capture, and meiotic mapping, combined with previously generated dideoxy shotgun sequence. New sequences were generated from 17th-generation individuals from the same inbred Nigerian line that was used in the original Sanger shotgun sequencing[45].

The new reference assembly, version 10 (v10), spans 1448.4 megabases (Mb) and is substantially more complete than the previous (v9) sequence[25], assigning 219.2 Mb more sequence to chromosomes (Supplementary Table 1). The v10 assembly is also far more contiguous, with half of the sequence contained in 32 contigs longer than 14.6 Mb (in comparison, this N50-length was. 71.0 kilobases [kb] in v9). The assembly captures 99.6% of known coding sequences (Supplementary Table 2 and Supplementary Note 2). We found that the fragmented quality of earlier assemblies was due, in part, to the fact that 68.3 Mb (4.71%) of the genome was not sampled by the 8× redundant Sanger dideoxy whole-genome shotgun dataset[45] (Supplementary Fig. 2a–c and Supplementary Note 2). These missing sequences are apparently due to non-uniformities in shotgun cloning and/or sequencing (Supplementary Fig. 2d–f). Previously absent sequences are distributed across 140.5k blocks of mean size 485.7 basepairs (bp) (longest 50.0 kb) on the new reference assembly, are enriched for sequences with high GC content (Supplementary Fig. 2g), and capture an additional 6774 protein-coding exons from among 4718 CDS sequences (Supplementary Fig. 2d, e). The enhanced contiguity of v10 is accounted for by the relatively uniform coverage of PacBio long-read sequences along the genome, as expected from other studies[46–49]. Most remaining gaps are in highly repetitive and satellite-rich centromeres and subtelomeric regions (see below) (Supplementary Fig. 2a).

### Additional chromosome-scale frog genomes

To assess the evolution of chromosome structure across a diverse set of frogs, we generated chromosome-scale genome assemblies for three new emerging model species, including the Zaire dwarf clawed frog *Hymenochirus boettgeri* (a member of the family Pipidae along with *Xenopus* spp.), and two neobatrachians: the Puerto Rican coquí *Eleutherodactylus coqui* (family Eleutherodactylidae) and the túngara frog *Engystomops pustulosus* (family Leptodactylidae). These chromosome-scale draft genomes were primarily assembled from short-read datasets and chromatin conformation capture (Hi-C) data (Supplementary Data 1, Supplementary Table 3, and Supplementary Note 3). To further expand the scope of our comparisons, we also updated the assemblies of two recently published frog genomes: the

African bullfrog *Pyxicephalus adspersus*[28], from the neobatrachian family Pyxicephalidae, and the Ailao moustache toad *Leptobrachium* (*Vibrissaphora*) *ailaonicum*[29], from the family Megophryidae (Supplementary Fig. 3 and Supplementary Note 3). These species span the pipanuran clade, which comprises all extant frogs except for a small number of phylogenetically basal taxa, such as *Bombina*[40] and *Ascaphus*[50].

The chromosome numbers of the new assemblies agree with previously described karyotypes for *E. coqui*[51] ($2n = 26$) and *E. pustulosus*[52] ($2n = 22$). The literature for *H. boettgeri*, however, is more equivocal, with reports[53,54] of $2n = 20–24$. The $n = 9$ chromosomes of our *H. boettgeri* assembly are consistent with our chromosome spreads (Supplementary Fig. 3a). The karyotype variability in the published literature and discrepancy with the karyotypes of our *H. boettgeri* samples may be the result of cryptic sub-populations within this species or segregating chromosome polymorphisms.

## Protein-coding gene set for *X. tropicalis*
The improved *X. tropicalis* genome encodes an estimated 25,016 protein-coding genes (Supplementary Table 4), which we predicted by taking advantage of 8580 full-length-insert *X. tropicalis* cDNAs from the "Mammalian" Gene Collection[55] (MGC), 1.27 million Sanger-sequenced expressed sequence tags[45] (ESTs), and 334.5 gigabases (Gb) of RNA-seq data from an aggregate of 16 conditions and tissues[56,57] (Supplementary Data 1 and Supplementary Note 2). The predicted gene set is a notable improvement on previous annotations, both in completeness and in full-length gene-level accuracy, due in part to the more complete and contiguous assembly (Supplementary Fig. 1, Supplementary Table 2, and Supplementary Note 2). In particular, single-molecule long reads filled gaps in the previous *X. tropicalis* genome assemblies that likely arose from cloning biases in the Sanger sequencing process, encompassing exons embedded in highly repetitive sequences (Supplementary Fig. 2).

A measure of this completeness and the utility of the *X. tropicalis* genome is provided by comparing its gene set with those of vertebrate model systems with reference-quality genomes, including chicken[58], zebrafish[59], mouse[60], and human[61,62] (Supplementary Fig. 4a–c). Notably, despite the closer phylogenetic relationship between birds and mammals, *X. tropicalis* shares more orthologous gene families (and mutual best hits) with human than does chicken, possibly because of the loss of genomic segments in the bird lineage[23,63] and/or residual incompleteness of the chicken reference sequence, due to the absence of several microchromosomes[58]. For example, of 13,008 vertebrate gene families with representation from at least four of the vertebrate reference species, only 341 are missing from *X. tropicalis* versus 1110 from chicken (Supplementary Fig. 4a). The current *X. tropicalis* genome assembly also resolves gene order and completeness of gene structures in the long subtelomeres that were missed in previous assemblies due to their highly repetitive nature (Supplementary Fig. 2).

## Protein-coding gene sets for additional frogs
We annotated the new genomes of *E. coqui*, *E. pustulosus*, *H. boettgeri*, and *P. adspersus* using transcriptome data from these species (Supplementary Data 1) and peptide homology with *X. tropicalis* (Supplementary Tables 5 and 6). To include mustache toad in our cross-frog comparisons, we adopted the published annotation from ref. 29 (Supplementary Note 3). We found 14,412 orthologous groups across the five genera with OrthoVenn2[64], including genes found in at least four of the five frog genera represented (Supplementary Fig. 4d). As expected, due to its reference-quality genome and well-studied transcriptome, only 72 of these clusters were not represented in *X. tropicalis* (and only 42 clusters from gene families present in six or more members among a larger set of seven frog species, see Supplementary Fig. 4e); the additional frog genomes each had between 575 and 712 of these genes missing (or mis-clustered), suggesting better than 95%

completeness in the other species. For analyses of synteny, we further restricted our attention to 7292 one-to-one gene orthologs that were present on chromosomes (as opposed to unlinked scaffolds) in the "core" genomes *X. tropicalis, H. boettgeri, E. coqui, E. pustulosus*, and *P. adspersus*. The total branch length in the pipanuran tree shown in Fig. 1 (including both *X. laevis* subgenomes) is 2.58 substitutions per four-fold synonymous site.

## Repetitive landscape
Centromeric and telomeric tandem repeats play a critical role in the stability of chromosome structure[65]. Nonetheless, other kinds of repeats also play a role in the preservation of these important chromosome landmarks[66, 67]. The new *X. tropicalis* v10 assembly captures sequences from centromeres and distal subtelomeres that were fragmented in the previous assemblies[25,45]. The percentage of the genome covered by transposable elements is slightly higher than previously reported[45] (36.82% vs. 34%) (Supplementary Table 7).

Insertional bias in the pericentromeric regions is observed for specific families of long interspersed elements (LINEs), including the relatively young Chicken Repeat 1 (CR1)[68] (3.14% of the genome) and the ancient L1 (1.06%) (Fig. 2 and Supplementary Fig. 5). The *X. tropicalis* v10 assembly captures significantly more tandem repeats in the distal subtelomeric portions of the genome relative to earlier assemblies. An exhaustive search for tandem repeats using Tandem Repeats Finder[69] determined that 10.67% of the chromosomes are covered by tandem arrays consisting of 5 or more monomeric units greater than 10 bp. Many tandem repeat footprints lie in the gaps of previous assemblies[25,45] (Supplementary Fig. 2). Our new hybrid genome assembly closed many gaps containing centromeric and subtelomeric tandem repeats, and captured numerous subtelomeric genes (Supplementary Fig. 2). The overall repeat landscape derived from the *X. tropicalis* assembly is mirrored in the other frog assemblies, with similar centromeric repeats, and lengthy subtelomeres, as discussed below.

## Genetic variation
The inbred *X. tropicalis* reference genotype was nominally derived from 17 generations of brother-sister mating, starting with two Nigerian founders. In the absence of selection, this process should lead to an increasingly homozygous genome due to increasing identity by descent of the two reference haplotypes, with residual heterozygosity confined to short blocks totaling a fraction ~$1.17 \times (0.809)^t$ of the genetic map[70], or 3.2% after $t = 17$ generations of full-sib mating. In contrast, we observe that 11.7% of the genome (125.12 cM out of a total of 1070.16 cM) exhibits residual heterozygosity (Supplementary Fig. 6). While this excess could be explained by balancing selection due to recessive lethals, a more mundane possibility is that some non-full-sib mating occurred during the inbreeding process. Errors early in the inbreeding process would be consistent with the unexpectedly high heterozygosity (~44%) observed in two 13th-generation members of the lineage (Supplementary Fig. 6), which far exceeds the 7.4% theoretical expectation from repeated full-sib mating. The approximately fourfold further reduction from these individuals to our 17th-generation reference, however, is consistent with theoretical expectations in the absence of selection.

Residual blocks of heterozygosity after inbreeding reflect distinct founder haplotypes. Within these blocks, we observe 3.0 single-nucleotide variants per kilobase, which serves as an estimate of the heterozygosity of the wild Nigerian population. To begin to develop a catalog of segregating variation in *X. tropicalis*, we also shotgun-sequenced pools of frogs from the Nigerian and Ivory Coast B populations, which are the two main sources of experimental animals. These two populations have been previously analyzed using SSLP markers[71]. From our light pool shotgun analysis, we identified a total of 6,546,379 SNPs, including 2,482,703 variants in the Nigerian pool and 4,661,928

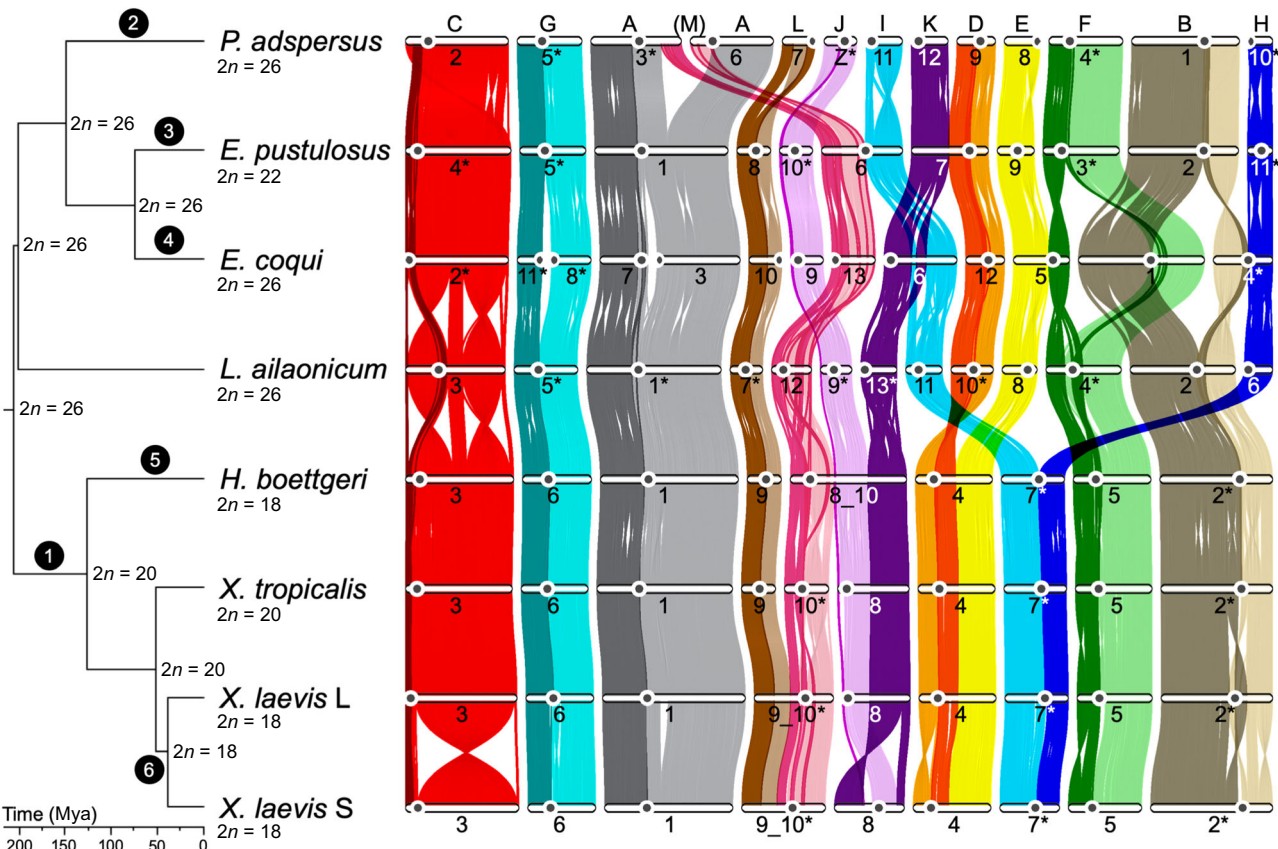

**Fig. 1 | Phylogenetic tree and gene ortholog alignment.** The phylogenetic tree of the seven analyzed species, calculated from fourfold degenerate sites and divergence time confidence intervals, drawn with FigTree (commit 901211e, https://github.com/rambaut/figtree): *Xenopus tropicalis*, *X. laevis*, and *Hymenochirus boettgeri* (Pipoidea: Pipidae); *Leptobrachium* (*Vibrissaphora*) *ailaonicum* (Pelobatoidea: Megophryidae); *Engystomops pustulosus* (Neobatrachia [Hyloidea]: Leptodactylidae), *Eleutherodactylus coqui* (Neobatrachia [Hyloidea]: Euleutherodactylidae); and *Pyxicephalus adspersus* (Neobatrachia [Ranoidea]: Pyxicephalidae). The ancestral karyotype is labeled at each node on the tree. Black circles with white text refer to chromosome changes summarized in Table 1. The alignment plot was generated with JCVI using the 7292 described chromosome one-to-one gene orthologs from OrthoVenn2, followed by manual filtering of single stray orthologs. The Hi-C-derived centromere position is represented with a black circle on each chromosome. Ancestral chromosomes (A to M) are labeled at the top of the alignment based on the corresponding region in *P. adspersus*. The alignments for each ancestral chromosome are colored uniquely, with those upstream and downstream of the *X. tropicalis* centromeric satellite repeat colored in dark and light shades of the ancestral chromosome color. Chromosomes labeled with asterisks are shown reverse complemented relative to their orientations in the genome assembly. Mya millions of years ago, *n* the haploid chromosome number. Source data are provided as a Source Data file.

in the Ivory Coast B pool, with 598,252 shared by both pools, suggesting differentiation between populations (Supplementary Fig. 6 and Supplementary Note 2).

**Conserved synteny and ancestral chromosomes**

Comparison of the chromosomal positions of orthologs across seven frog genomes reveals extensive conservation of synteny and collinearity (Fig. 1 and Supplementary Fig. 7a–g). We identified 13 conserved pipanuran syntenic units that we denote A through M ("Methods" and Supplementary Note 4). Each unit likely represents an ancestral pipanuran chromosome, an observation consistent with the $2n = 26$ ancestral karyotype inferred from cytogenetic comparisons across frogs[36,72]. Over 95% (6952 of 7292) of chromosomal one-to-one gene orthologs are maintained in the same unit across the five frog species, attesting to the stability of these chromosomal elements (Fig. 1). The conservation of gene content per element is comparable to the 95% ortholog maintenance in the Muller elements in *Drosophila* spp[73]. Despite an over twofold difference in total genome size across the sampled genomes, each ancestral pipanuran element accounts for a nearly constant proportion of the total genome size, gene count, and repeat count in each species, implying uniform expansions and contractions during the history of the clade (Supplementary Fig. 7h).

At least some of these pipanuran elements have a deeper ancestry within amphibians. For example, the chromosomes of the discoglossid frog *Bombina variegata* ($n = 12$), an outgroup to the pipanurans, show considerable conservation of synteny with *X. tropicalis* based on linkage mapping[40]. Compared with the pipanuran ancestral elements described here, the nine *B. variegata* chromosomes 2, 3, 4, 5, 6, 8, 9, 10, and 12 correspond to nine pipanuran elements A, B, C, F, G, H, I, E, and J, respectively, extending these syntenic elements to the last common ancestor of *Bombina*+pipanurans (which does not have a common name). The remaining three *B. variegata* chromosomes 1, 7, and 11 are combinations of the remaining four pipanuran elements D, K, L, and M. Similarly, the genome of the axolotl, *Ambystoma mexicanum*, a member of the order Caudata (salamanders and newts) and ~292 million years divergent from pipanurans[74], also conserves multiple syntenic units with pipanurans (Supplementary Fig. 7i). For example, axolotl chromosomes 4, 6, 7, and 14 are in near 1:1 correspondence with pipanuran elements F, A, B, and K, respectively, although small pieces of F and A can be found on axolotl 10, and parts of B can be found on axolotl 9 and 13. Other axolotl chromosomes are fusions of parts of two or more pipanuran elements. For example, axolotl chromosome 5 is a fusion of a portion of J with most of G; the remainder of G is fused with a portion of L on the q arm of axolotl chromosome 2. Further

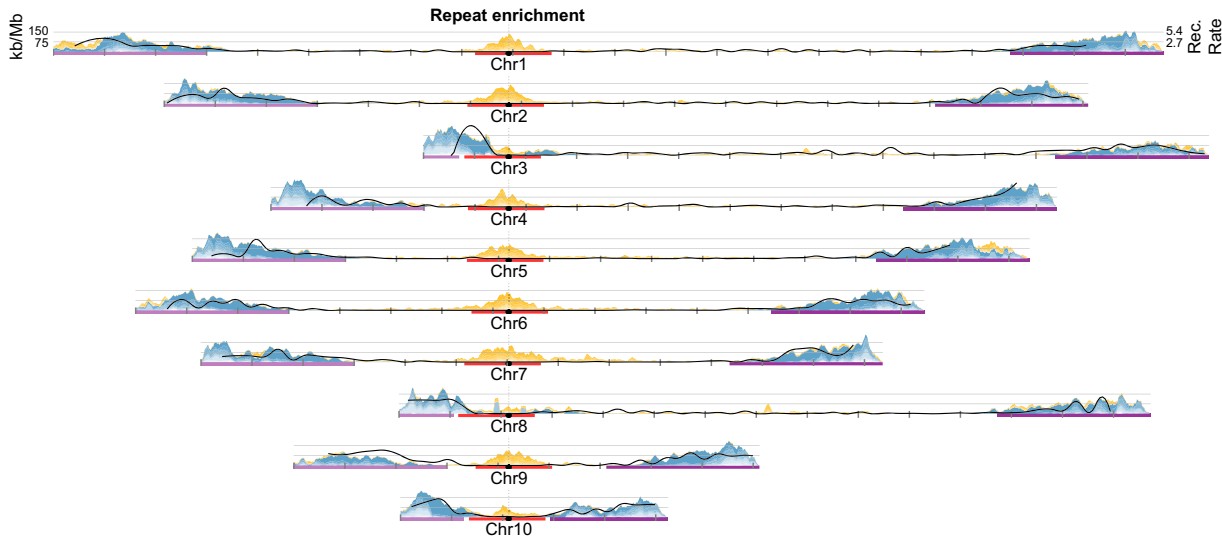

**Fig. 2 | Density of pericentromeric and subtelomeric repeats in *Xenopus tropicalis*.** Pericentromeric (red) and subtelomeric (purple) regions were used to obtain enriched repeats, excluding chromosomes with short p-arms (chromosomes 3, 8, and 10). Pericentromeric repeats (yellow) correspond to selected subsets of non-LTR retrotransposons (CR1, L1, and Penelope), LTR retrotransposons (Ty3), and DNA transposons (PiggyBac and Harbinger). Subtelomere-enriched repeats (blue) correspond mainly to satellite repeats and LTR retrotransposons (Ty3, Ngaro). Densities of each repeat type plotted as kb/Mb. Chromosomes are centered by the position of centromeric tandem repeats (black dots). Rates of recombination (Rec. rate) in cM/Mb are shown as solid black lines. Tick marks indicate 10 Mb blocks (Supplementary Fig. 5). kb kilobases, Mb megabases, cM centiMorgans. Source data are provided as a Source Data file.

comparisons are needed to determine which of these rearrangements occurred on the axolotl vs. the stem pipanuran lineage. Genomes from the superfamilies Leiopelmatoidea and Alytoidea, which diverged prior to the radiation of pipanurans, will also be informative.

Chromosomal conserved synteny across pipanuran frogs is comparable to that observed in birds, which have evolved by limited intra-chromosomal rearrangement from an $n = 40$ ancestor[43], mostly involving fusion of microchromosomes, as we find here for pipanurans (see below). The relative stasis of frog and bird chromosomes is in contrast to the variable karyotypes of mammals, which was first noted by Bush et al.[37] and is now extensively documented at the level of chromosomal painting[22] and genome sequence[42]. The reasons for these different modes of evolution remain unclear but are likely related to the difficulty in fixing partial-arm chromosomal rearrangements in large historically panmictic populations due to reduced fertility in translocation heterozygotes, as first noted by Wright[75]. Partial-arm rearrangements, as observed in mammals, can become fixed in populations that are dynamically subdivided by local extinction and colonization, which allows the reduced fertility of translocation heterozygotes to be overcome by genetic drift[76].

**Chromosome evolution**
Block rearrangements of the 13 ancestral elements dominate the evolutionary dynamics of pipanuran karyotypes (Table 1 and Fig. 1). While element C has remained intact as a single chromosome across the group (except for internal inversions), all of the other elements have experienced translocations during pipanuran evolution. During these translocations, the elements have remained intact except for the breakage of elements A and M by reciprocal partial-arm exchange observed in *P. adspersus* chromosomes 3 and 6.

To trace the evolutionary history of centromeres shown in Fig. 1, we inferred their positions using Hi-C contact map patterns, as in *X. tropicalis* (where centromeres were also confirmed by analysis of Cenp-a binding as described below). In general, the pericentromeres of other pipanurans were characterized by the same repetitive element families found in *Xenopus*, further corroborating their identification. Overall, we found broad pericentromeric conservation among the species analyzed (Figs. 1 and 3a).

Robertsonian or centric translocations involving breaks and joins near centromeres account for several of the rare rearrangements (Figs. 1 and 3b). For example, element G clearly experienced centric fission in the *E. coqui* lineage. Conversely, I and M underwent centric fusion in the *E. pustulosus* lineage. *E. coqui* has experienced the most intense rearrangement, including Robertsonian fissions of A and G, a Robertsonian fusion of I/K, and a significant series of Robertsonian rearrangements involving B, E, F, and H that resulted in Bprox/H, Bdist/Fdist, and E/Fprox (Table 1 and Supplementary Table 8). (Mechanistically, these "fissions" and "fusions" likely occur by translocations; see ref. 77 for a discussion.) Elements I and H form the two arms of a submetacentric chromosome in pipids (Fig. 3a), and therefore the pipid ancestor, but are found as either independent acrocentric chromosomes (e.g., in *P. adspersus* and *L. ailaonicum*) or as arms of

**Table 1 | Organization and conservation of the 13 ancestral chromosomes of pipanuran genomes**

| Phylogenetic position | Structural event |
|---|---|
| (1) Stem pipid lineage | J + K → JK |
| | D. + E. → D.E |
| | I• + •H → I • H (Rob. fusion) |
| (2) *P. adspersus* lineage after divergence from *R. temporaria* | A + M → A1.m1 + m2.A2 |
| (3) *E. pustulosus* lineage after divergence from *E. coqui* | M + I → M.I (Rob) |
| | K + D → K.D (Possible end-end) |
| (4) *E. coqui* lineage after divergence from *E. pustulosus* | G1 • G2 → G1• + •G2 (Rob. fission) |
| | A1 • A2 → A1• + •A2 (Rob. fission) |
| | I + K → I • K (Rob. fusion + inversion) |
| | E + F1•F2 + B1•B2 + H → E•F1 + F2•B2 + B1•H |
| (5) *H. boettgeri* lineage after divergence from *Xenopus* | M + J•K → MJK |
| (6) *X. laevis* progenitor lineage after divergence from *X. tropicalis* | L + M → LM |

*Rob* Robertsonian.
Middle-dots (i.e., "•") represent centromeres. Periods (i.e., ".") represent translocation breakpoints.

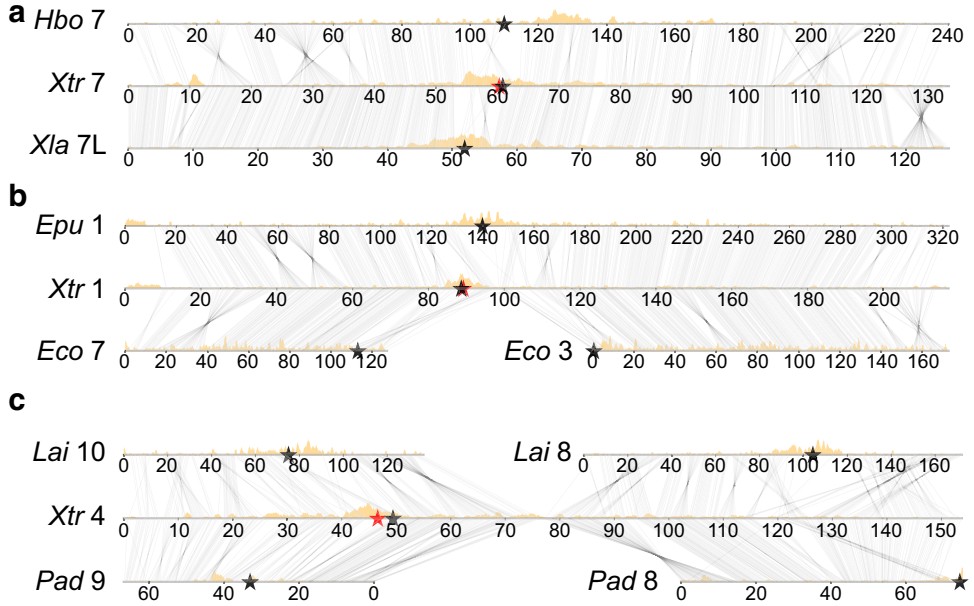

**Fig. 3 | Subtelomeric repeats highlight regions of chromosome fusion.** Examples of (**a**) conserved structure and pericentromere maintenance of *H. boettgeri* (Hbo), *X. tropicalis* (Xtr), and *X. laevis* (Xla) chromosomes; **b** a Robertsonian translocation in the lineage leading to *E. coqui* (Eco), shown compared with *E. pustulosus* (Epu) and *X. tropicalis*; and **c** an end-to-end fusion that occurred in the lineage giving rise to *X. tropicalis* and subsequent pericentromere loss, shown compared with *L. ailaonicum* (Lai) and *P. adspersus* (Pad). The analyzed species were visualized with a custom script, alignment_plots.py (v1.0, https://github.com/ abmudd/Assembly). For each plot, the Hi-C inference-based centromeric regions are depicted with black stars, the *X. tropicalis* centromeric satellite repeat from tandem repeat analysis with a red star (on *X. tropicalis* chromosomes 7 and 1 (**a**, **b**), the stars overlap), the density of L1 repeats per chromosome with gold densities, and the runs of collinearity containing at least one kilobase of aligned sequence between the species with connecting black lines. kb kilobases, Mb megabases. Source data are provided as a Source Data file.

(sub)metacentrics formed by centric fusion with other elements (Supplementary Table 8).

We also observed end-to-end "fusions"[78] of (sub)metacentric chromosomes, for example, the joining of D with K in *E. pustulosus*, and with element E in the common ancestor of pipids (*Hymenochirus* and *Xenopus*) (Figs. 1 and 3c). Since bicentric chromosomes are not stably propagated through mitosis, one of the two ancestral centromeres brought together by end-to-end fusion must be lost or inactivated, as shown in Fig. 3c for the ancient D–E fusion in pipids. We note that the D centromere persists in both end-to-end fusions involving D, suggesting that centromeres derived from different ancestral elements may be differentially susceptible to silencing, although with only two examples this could have happened by chance.

Using the pericentromeric and subtelomeric repeats landscape as a proxy, we found several examples of end-to-end chromosome fusions in which residual subtelomeric signals are preserved near the presumptive junctions (Fig. 3 and Supplementary Fig. 8). These include the end-to-end fusion of *X. tropicalis*-like chromosomes 9 and 10 (elements L and M) to produce the *X. laevis* chromosome 9_10 progenitor that is found in both the L and S subgenomes of this allotetraploid[27]. These *X. laevis* chromosomes display evidence of decaying subtelomeric signatures in the region surrounding the ancestral L–M fusion (Fig. 1 and Supplementary Fig. 8a, b). Similarly, enrichment of subtelomerically-associated repeats is observed in *H. boettgeri* chromosome 8_10 (Supplementary Fig. 8c–e) near the junction between the portions of the chromosome with M and J/K ancestry (the J/K fusion occurred near the base of pipids). In both cases, the centromere from element M (i.e., the centromere in *X. tropicalis* chromosome 9) is maintained after fusion. The inversion of the p-arm from chromosome 8S also has evidence of decaying sequence but the median is less than the median Jukes-Cantor (JC) distance at the chromosome 9_10 fusion, suggesting that the fusion preceded the inversion.

## Rate of karyotype change

The long-range and, in most cases, chromosome-scale collinearity (Supplementary Fig. 7 and Supplementary Table 9) among the frog species we examined, despite a combined branch length of 1.05 billion years (Supplementary Tables 10 and 11), parallels the conserved synteny observed in birds[79] and reptiles[80], but differs from the substantial chromosome variation found in mammals[22,41]. Maintenance of collinear blocks may reflect an intrinsically slow rate of rearrangement in frogs, perhaps a consequence of large regions devoid of recombination, or selection favoring retention of specific gene order and chromosome structure related to chromosomal functions. We inferred 8 fusions, 2 fissions, one pairwise, and one four-way reciprocal fusion; counting the last as a composite of three pairwise rearrangements yields a total of 17 translocations (excluding smaller intra-chromosome rearrangements) corresponding to an average rate of one karyotype change every 62 million years (Fig. 1 and Table 1). This rate is similar to the rate of one chromosome number change every 70 to 90 million years as previously proposed for frogs and some mammals[33,37] but still slower than karyotype change rates for most mammals[81] and many reptiles[82]. Of course, our rate calculation is based on only seven species, and the rate may vary depending on the species analyzed. Some frog taxa, such as *Eleutherodactylus* spp. (2n = 16–32) and *Pristimantis* spp[51]. (2n = 22–38), have experienced higher rates of karyotype change. On the other hand, other lineages, such as those leading to *Leptobrachium ailaonicum*, *L. leishanense*[14], and *Rana temporaria*[83], have had no detectable inter-chromosome exchange over the past 205 million years (Fig. 1). Nonetheless, this analysis of chromosome variation across the frog lineage is consistent with an overall slow rate of karyotype evolution[84].

Considering rearrangement rate variation across taxa, we can ask whether any of the individual branches show an unusually high or low number of translocations relative to the overall pipanuran rate. The absolute karyotype stasis of *L. ailaonicum* over ~200 My is only

marginally slower than the pipanuran average (two-sided test, $P = 0.04$ under a simple Poisson model of 1 change every 62 My, before family-wise correction for testing of multiple lineages). Conversely, the *E. coqui* lineage has experienced six translocations during a time interval in which only one rearrangement would be expected. This is a significant enrichment relative to the Poisson model ($P = 1 \times 10^{-3}$) and is the only branch on which the constant rate hypothesis is rejected. Notably, *Euleutherodactylus* is the most karyotypically variable frog genus, suggesting possible ongoing karyotypic instability[84,85].

Regarding chromosome stability, our collection only includes one example in which a chromosome arm is disrupted by translocation; all other changes are either Robertsonian (involving breaks near a centromere) or end-to-end (near a telomere). This observation allows us to reject ($P < 4 \times 10^{-4}$) a simple random break model, under which we would expect ~12.3 chromosome arms to be broken across our phylogeny (Supplementary Note 4). This suggests that centromeric and telomeric regions are more prone to breakage, and/or breaks within chromosome arms are selected against. The latter model is consistent with a reduced probability of fixation of reciprocal (partial-arm) translocations due to selection against reduced fertility in heterozygotes[75], which can be overcome by genetic drift under some conditions[76].

## Centromeres, satellites, and pericentromeric repeats

The stasis of *Xenopus* chromosomes relative to other frogs (see above) allows us to examine the repetitive landscape of chromosomes that are not frequently rearranged by translocation and may be approaching a structural equilibrium. Vertebrate centromeres are typically characterized by tandem families of centromeric satellites (e.g., the alpha satellites of humans) that bind to the centromeric histone H3 protein, Cenp-a, a centromere-specific variant of histone H3[65,86]. Cenp-a binding satellites have been described in *X. laevis*[87], and here we find distantly related *X. tropicalis* satellite sequences that also co-precipitate with Cenp-a. Thus, chromatin immunoprecipitation and sequencing (ChIP-seq) shows that Cenp-a binding coincides with the predictions of centromere positions derived from chromatin conformation analysis and repetitive content (Supplementary Figs. 5a–c and 9a–c and Supplementary Tables 12 and 13). Importantly, this concordance supports the prediction of centromere position for other species that we infer below. The Cenp-a-bound sequences are arrays of 205-bp monomers that share a mean sequence identity greater than 95% at the nucleotide level, with a specific segment of the repeating unit showing the greatest variability (Supplementary Fig. 9d, e). The *X. tropicalis* centromere sequence is different from centromeric-associated repeats found in *X. laevis*[87,88], suggesting the sequences evolve rapidly after speciation but are maintained across chromosomes within the species.

All pericentromeric regions of (sub)metacentric *X. tropicalis* chromosomes are enriched in retrotransposable repetitive elements (15 Mb regions shown in Fig. 2). In other vertebrate species and *Drosophila*, retrotransposable elements from the pericentromeric regions are involved in the recruitment of constitutive heterochromatin components[89,90]. Among the pericentromerically-enriched repeats we identified specific families belonging to LTR retrotransposons (Ty3), non-LTR retrotransposons (CR1, Penelope, and L1), and DNA transposable elements (PIF-Harbinger and piggyBac families) (Fig. 2 and Supplementary Fig. 5). CR1 (CR1-2_XT) is the most prevalent and among the youngest of all pericentromeric retrotransposons (mean Jukes-Cantor (JC) distance to consensus of 0.05). In contrast, L1 and Penelope types have a mean JC greater than 0.4 (Supplementary Fig. 5). The age of the repeats, indirectly measured by the JC distance, suggests that pericentromeric retrotransposons have experienced different bursts of activity and tendency to insert near the centromere. Expression of active retrotransposons and random insertion can compromise chromosome stability, and because silencing of these is crucial, genomes develop mechanisms to rapidly silence them. Such insertions may be positively selected, and therefore amplified, to establish pericentromeric heterochromatin, but may be counter-selected when they insert in gene-rich chromosome arms.

## Recombination and extended subtelomeres

With chromosome sequences in hand, we studied the distribution of recombination along *X. tropicalis* chromosomes using a previously generated Nigerian-Ivory Coast $F_2$ cross[25] (Supplementary Note 5 and Supplementary Data 2). Half of the observed recombination is concentrated in only 160 Mb (11.0% of the genome) and 90% of the observed recombination occurs in 540 Mb (37.3%). In contrast, the extended central regions of each chromosome are "cold," with recombination rates below 0.5 cM/Mb and that are often indistinguishable from zero in our data (Supplementary Fig. 10a, b and Supplementary Table 14). Strikingly, we find that (sex-averaged) recombination is concentrated within just 30 Mb of the ends of each chromosome and occurs only rarely elsewhere (Supplementary Fig. 10a). The regions of the subtelomeres experiencing high recombination are nearly sixfold longer than in non-amphibian genomes[91,92]. The rates of recombination in *Xenopus* subtelomeres were not previously determined, since the repeat-rich subtelomeres were absent from earlier assemblies, and markers present in those regions showed insufficient linkage to be incorporated into linkage maps[25].

Elevated rates of recombination near telomeres and long central regions of low recombination have been observed in the macro-chromosomes of diverse tetrapods, including birds[92,93], snakes[94], and mammals[95–97]. This pattern appears to be independent of the involvement of the chromatin modifier PRDM9 in defining recombination hotspots[98] since dogs lack PRDM9 but show the same pattern, with elevated recombination in promoter regions and around CpG islands[96]. Conversely, snakes possess the *prdm9* gene but also show hotspots of recombination concentrated in promoters and functional regions[94]. Since amphibians lack the *prdm9* gene[99], we further analyzed the genomic features that colocalized in subtelomeric regions prone to recombination.

To assess sequence features associated with enriched recombination, we focused on the extended subtelomeres, defined as the terminal 30 Mb of all (sub)metacentric chromosomes and the terminal 30 Mb excluding the 15 Mb surrounding the pericentromeric regions of acrocentric chromosomes (3, 8, and 10) (Fig. 2). The median recombination rate in the extended subtelomeres (1.72 cM/Mb) is over tenfold higher than the median rate observed in the rest of the chromosome arms (0.14 cM/Mb) (two-sample Kolmogorov–Smirnov test, two-sided, Hochberg-corrected $P = 5.2 \times 10^{-321}$) (Supplementary Fig. 10c and Supplementary Note 5). The recombination rate in the 5-Mb region surrounding the centromeric tandem repeats is even lower (0.01 cM/Mb). Since constitutive heterochromatin in pericentromeric regions is known to repress recombination, this observation is expected (reviewed in refs. 100,101). However, the centromeres of acrocentric chromosomes lie within 30 Mb of telomeres and preclude the presence of extended subtelomere-associated repeats (Fig. 2 and Supplementary Fig. 11).

We examined the relationship between rates of recombination against repetitive elements and sequence motifs associated with recombination hotspots in other vertebrate species (Supplementary Fig. 12a and Supplementary Table 14). Similar to chicken and zebra finch, recombination is the highest in subtelomeres and positively correlates with GC content[92,93,102], which is consistent with GC-biased gene conversion[83,103,104] in recombinogenic regions (median GC = 42.5% in the 74 Mb in which half of the recombination occurs) vs. the non-recombinogenic centers of chromosomes (median 38.8%). As in zebra finch (Supplementary Fig. 13), recombination in *X. tropicalis* is strongly correlated with satellite repeats (Pearson's correlation, $r = 0.68$, $R^2 = 0.457$). The high density of satellite repeats (Supplementary Table 15) in highly recombinogenic subtelomeric regions suggests that

unequal crossing over during meiotic recombination mediates tandem repeat expansions[105,106]. Notably, in the extended subtelomeric regions tandem repeats are enriched in specific tetrameric sequences (TGGG, AGGG, and ACAG) compared to non-tandem repeats (Supplementary Fig. 12b). In contrast, centromeric tandem repeats are completely devoid of these short sequences.

Some of the tandem arrays enriched in the terminal 30 Mb of all chromosomes derive from portions of transposable elements, such as SINE/tRNA-V, LINE/CR1, DNA/Kolobok-2 (Supplementary Fig. 11 and Supplementary Table 16). For example, the minisatellite expansion that arose from the family of SINE/tRNA-V present in the pipid lineage[107] amplified a 52-bp portion of the 3'UTR-tail from the SINE/tRNA-V element in *Xenopus tropicalis* and other frog species (Supplementary Table 17). Although intact SINE/tRNA-V elements are distributed throughout the genome, the minisatellite fragment is only expanded in subtelomeric SINE/tRNA-Vs, suggesting that recombination in subtelomeres has driven minisatellite expansion (Supplementary Figs. 11 and 14). Interestingly, although the satellite expansions are similar in *X. laevis* and *X. tropicalis*, they differ in other frogs, suggesting that different satellite expansions can occur repeatedly during the maintenance of the long subtelomeric regions (see below).

We hypothesize that the high rate of recombination in the extended subtelomeres of frog chromosomes drives tandem repeat expansion through illegitimate homologous recombination and, in the process, increases GC content (Supplementary Fig. 14d, e). Unfortunately, it is difficult to resolve cause and effect with observational data,

and we cannot rule out the alternative hypothesis that meiotic recombination is promoted by preferential DNA breakage at short sequence motifs (Supplementary Fig. 12b), which is then repaired by homologous recombination.

## Chromatin conformation correlates with cytogenetic features

To further refine our understanding of chromosome structure in *X. tropicalis*, we studied chromatin conformation capture ("Hi-C") data from nucleated blood cells. These experiments link short reads representing sequences in close three-dimensional proximity[108]. Figure 4 shows mapped Hi-C read pairs for chromosomes 1 and 2, with different minimum mapping quality thresholds above and below the diagonal (Supplementary Fig. 1e and Supplementary Note 5). We consistently observe a "wing" of intra-chromosome contacts transverse to the main diagonal, which (1) intersects the main diagonal near the cytogenetically defined Cenp-a-binding centromere, and (2) indicates contacts between p and q-arms (Supplementary Figs. 1e and 15). These observations imply that interphase chromosomes are "folded" at their centromeres, with contacts between distal arms. We also observe enriched inter-chromosome contacts among centromeres and among chromosome arms along a centromere-to-telomere axis, suggesting that chromosomes are organized in a polarized arrangement in the nucleus (Supplementary Figs. 9a and 15 and Supplementary Table 18). Notably, the correlation between centromere position and the observed intra-chromosome folding and inter-chromosome contacts at centromeres allows us to use Hi-C analysis and principal

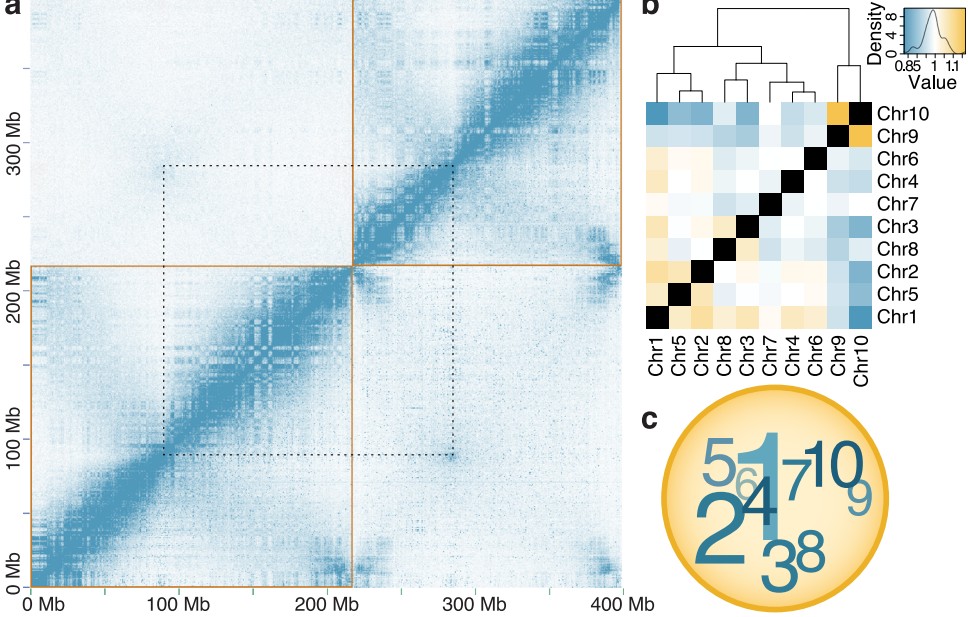

**Fig. 4 | Organization of *X. tropicalis* chromosomes into Rabl-like configuration and distinct nuclear territories. a** Hi-C contact matrices for chromosomes 1 and 2 (lower-left and upper-right gold boxes, respectively) showing features of the three-dimensional chromatin architecture within *X. tropicalis* blood cell nuclei. Blue pixels represent chromatin contacts between *X–Y* pairs of 500 kb genomic loci, with intensity proportional to contact frequency. Hi-C read pairs are mapped stringently (MQ ≥ 30) above the diagonal and permissively (MQ ≥ 0) below the diagonal. The characteristic A/B-compartment ("checkerboard") and Rabl-like ("angel wing") interarm contact patterns within each chromosome are evident. Above the diagonal, an increased frequency of interchromosomal chromatin contacts is observed between pericentromeres (connected by dotted lines) and between chromosome arms (Supplementary Tables 18, 19, and 21), suggesting a centromere-clustered organization of chromosomes in a Rabl-like configuration. Below the diagonal, high-intensity pixels near the ends of chromosomes not present above the diagonal suggest a telomere-proximal spatial bias in the

distributions of similar genomic repeats. See Supplementary Fig. 1e for a plot showing all chromosomes. **b** Chromosome territories within the nucleus. Yellow, white, and blue colors indicate the normalized relative enrichment, parity, and depletion of chromatin contacts between non-homologous chromosomes (Supplementary Tables 21 and 22). For example, chromosome 1 exhibits higher relative contact frequencies with all chromosomes except chromosomes 7, 9, and 10, which are generally depleted of contacts except among themselves (MQ ≥ 30; $\chi^2$ (81, $n = 24,987,749$) = 3,049,787; Hochberg-corrected $P < 4.46 \times 10^{-308}$; Relative range: 0.82774–1.16834). Note, due to the inbred nature of the Nigerian strain, contacts could not be partitioned by haplotype, and so the results reported here represent chromosomal averages. **c** Schematic representation of chromosome territories from (**b**). The size of each chromosome number is approximately proportional to the number of enriched interactions. Darker and lighter colors indicate chromosomes nearer and more distant to the reader, respectively. Mb megabases, MQ mapping quality. Source data are provided as a Source Data file.

component analysis (PCA) of intra- and inter-chromosome contacts[109] to infer the likely centromeric positions based purely on Hi-C data in frogs whose cytogenetics are less well-studied (see below).

Taken together, these intra- and inter-chromosome contacts in *Xenopus* blood cells are consistent with a Rabl-like (Type-I[110]) chromosome configuration[111,112]. Such associations among centromeres and among telomeres, first observed in salamander embryos[111], have been observed in other animals[110,113–117], fungi[110,118,119], and plants[109,110,120–122]. Outside of mammals, Rabl-like contacts have been observed in a wide diversity of taxa. Hoencamp et al.[110]. surveyed 24 plant and animal species using Hi-C and observed Rabl-like patterns in 14 (58.3%) of them. Out of seven vertebrates sampled, however, only *Xenopus laevis* fibroblasts showed a Rabl-like pattern. We note that Hi-C patterns can depend on cell type, cell cycle stage, and developmental time; and while Rabl-like Hi-C patterns are often absent from tissue samples used in mammalian genome sequencing projects, they have been observed in studies of mouse and human cell lines (Supplementary Note 5).

In *X. tropicalis*, this configuration is understood to be a relict structure from the previous mitosis[123,124] in which the chromosomes have become elongated and telomeres clustered on the inner nuclear periphery. Dernburg and colleagues[125] reasoned that the Rabl configuration observed in *Drosophila* embryonic nuclei[126,127] is a result of anaphase chromosome movement and, due to their rapidly dividing nature, such chromosomes are unable to "relax" into a diffused chromatin state. Consistent with this, we find that Rabl-like chromosomal interarm contacts in early frog development (NF stages 8–23) appear more tightly constrained (mean ± SEM: sum of squared distances [SSD] 1.384 ± 0.066, centromere-to-telomere-polar interarm contact enrichment [CTP] 2.492 ± 0.179) in these rapidly dividing cells. Notably, more specialized (liver and brain) *X. tropicalis* adult tissues, except for blood cell nuclei (SSD 1.465, CTP 1.813), show less chromosomal interarm constraint (mean ± SEM: SSD 5.233 ± 1.258, CTP 1.362 ± 0.153) (Supplementary Fig. 16, Supplementary Table 19, and Supplementary Note 5). Although it is possible that some amount of Hi-C signal may be due to residual incompleteness in the assembly and concomitant mismapping of reads to repeat sequences, these observations are robust to quality filtering, even when using single-copy sequences. Furthermore, such contacts are similarly weak in sperm cells[16] (SSD 6.285, CTP 1.056), a control that argues strongly against sequence mismapping artifacts (Supplementary Note 5). As noted above, the presence and strength of Rabl-like configurations vary depending on the tissue, cell type, and developmental time. Such variability highlights the need to sample a broader diversity of tissues and time points to characterize completely the Rabl-like chromosome structures in *X. tropicalis*.

## Chromatin compartments

Chromatin contacts in human[108,128,129], mouse[129], chicken[130] and other phylogenetically diverse species[131–133] often show a characteristic checkerboard pattern that is superimposed on the predominant near-diagonal signal. This pattern implies an alternating A/B-compartment structure with enriched intra-compartment contacts within chromosomes (Fig. 5a), which has been linked with G-banding in humans[134]. *X. tropicalis* also exhibits an A/B-compartment pattern, which emerges as alternating gene-rich ("A") and gene-poor ("B") regions (median 19.99 genes/Mb and 9.99 genes/Mb, respectively) (Fig. 5b). Despite their twofold difference in gene content, A and B-compartment lengths are comparable, with approximately exponential distributions (Supplementary Fig. 17). The arithmetic mean sizes are A = 1.32 Mb, B = 1.48 Mb; the corresponding geometric means (i.e., the exponential of the arithmetic mean of logarithms of lengths) are somewhat shorter (A = 0.807 Mb, B = 0.946 Mb). A/B compartments are also differentiated by repetitive content[129], with A-compartment domains showing slight enrichment (1.21–1.44-fold) in DNA transposons of the DNA/Kolobok-T2, DNA/hAT-Charlie, and Mariner-Tc1 families.

B-compartment domains had significantly higher enrichment for DNA transposons (DNA/hAT-Ac, Mar-Tigger) and retrotransposons (Ty3/metaviridae and CR1), among other repeats (1.12–2.11-fold) (Fig. 5c, Supplementary Table 20). The association between repeats over-represented in A and B compartments is also captured in one of the principal components obtained from the repeat densities of all chromosomes (Supplementary Note 5); we detect a modest negative correlation (Pearson's $r = -0.44$) between A/B compartments and the third principal component obtained from the repeat density matrix (Supplementary Fig. 5b). The association between chromatin condensation and repeat type could be due to a preference for certain transposable elements to insert in specific chromatin contexts, or chromatin condensation to be controlled, in part, by transposable element content, or a combination of these factors. However, we were unable to find any correlation of A/B compartments with the G-banding of condensed chromosomes in *X. tropicalis*[135,136].

## Higher-order chromatin interactions

Chromatin conformation contacts also provide clues to the organization of chromosomes within the nucleus. We observe non-random ($x^2$ (81, $n = 24{,}987{,}749$) = 3,049,787; Hochberg-corrected $P < 4.46 \times 10^{-308}$) associations between chromosomes in blood cell nuclei (Fig. 4b and Supplementary Tables 21 and 22): (a) chromosome 1 is enriched for contacts with chromosomes 2–8 (mean 1.05× enrichment), and depleted of contacts with 9 and 10 (mean 0.89×); (b) among themselves, chromosomes 2–8 show differential contact enrichment or depletion; and (c) chromosomes 9 and 10 are enriched (1.17×) for contacts with one another, but are depleted of contacts with all other chromosomes. These observations suggest the presence of distinct chromosome territories[111,137–139], where chromosomes 2–8 are localized more proximal to—and arrayed around—chromosome 1, with chromosomes 9 and 10 relatively sequestered from chromosome 1 (Fig. 4c). The contact enrichment between chromosomes 9 and 10 is particularly notable because these short chromosomes (91.2 and 52.4 Mb, respectively) have become fused in the *X. laevis* lineage[140], which might have been enabled by their persistent nuclear proximity[141–143].

Between chromosomes, p-p and q-q arm interactions exhibit a small but significant enrichment (1.059× enrichment; $x^2$ (1, $n = 24{,}786{,}496$) = 17,037; Hochberg-corrected $P < 4.46 \times 10^{-308}$) over p-q arm contacts. This is a general feature of (sub)metacentric chromosomes observed in other frog genomes (Supplementary Table 21), except *E. coqui* (0.928× enrichment; $x^2$ (1, $n = 6{,}850{,}547$) = 3,914; Hochberg-corrected $P < 4.46 \times 10^{-308}$), the chromosomes of which appear predominantly acrocentric or telocentric. Finally, the p-arms of chromosomes 3, 4, 8, and 9 are enriched for contacts with both p and q-arms of chromosome 10, with the acrocentric chromosomes 3 and 8 showing the strongest relative enrichment and a slight preference between p-arms. The q-arms of chromosomes 3 and 8, however, exhibit a slight enrichment for contacts with the larger (sub)metacentric chromosomes 1, 2, 4, and 5. Taken together, these observations suggest possible colocalization of the p and q-arms of chromosomes 3 and 8 in *X. tropicalis* blood cell nuclei.

## Future impacts

Anuran amphibians play a central role in biology, not simply as a globally distributed animal group, but also as key subjects for research in areas that range from ecology and evolution to cell and developmental biology. The genomic resources generated here will thus provide important tools for further studies. Given the crucial role of *X. tropicalis* for genomic analysis of development and regeneration[144,145], the improvements to our understanding of its genome reported here will provide a more finely-grained view of biomedically important genetic and epigenetic mechanisms. This new genome is also important from the standpoint of evolutionary genomics, as comparisons between the genomes of *X. tropicalis* and *X. laevis* shed light on the

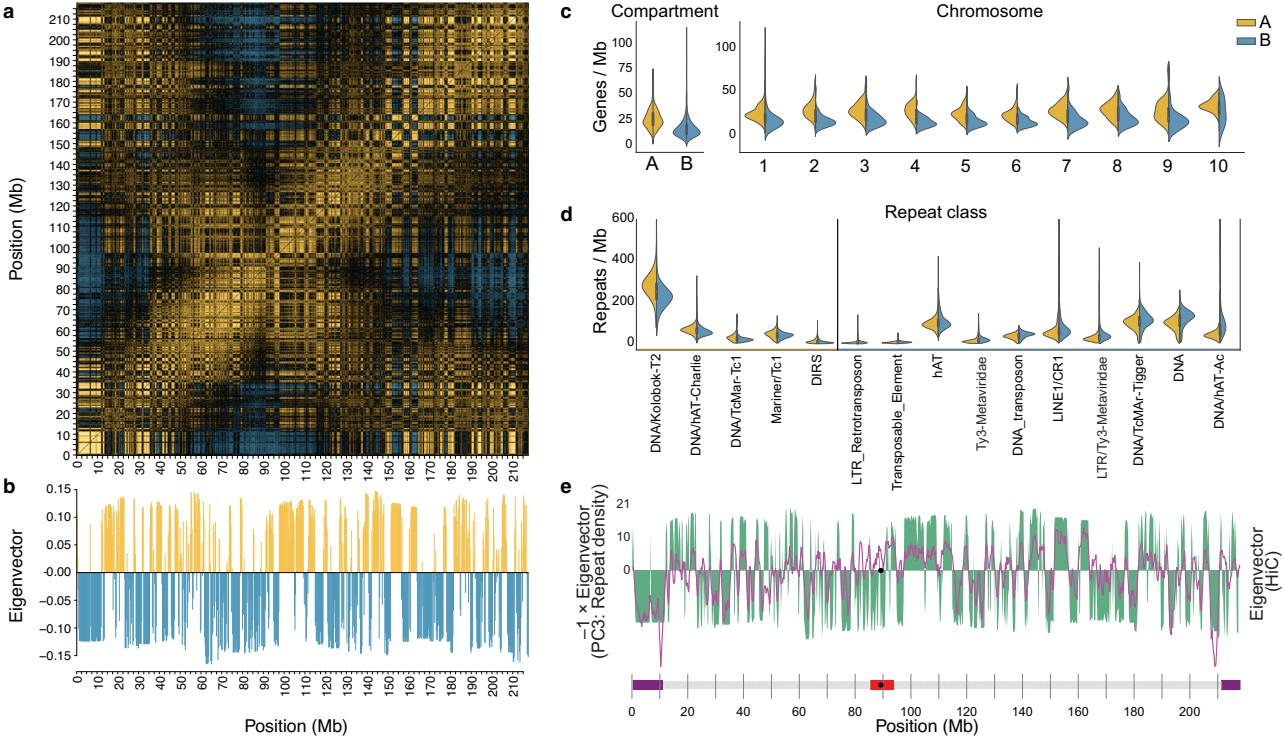

**Fig. 5 | A/B-compartment structure and gene/repeat densities. a** Correlation matrix of intra-chromosomal Hi-C contact densities between all pairs of non-overlapping 250 kb loci on chromosome 1. Yellow and blue pixels indicate correlation and anti-correlation, respectively, and reveal which genomic loci occupy the same or different chromatin compartment. Black pixels indicate weak/no correlation. **b** The first principal component (PC) vector revealing the compartment structure along chromosome 1, obtained by singular value decomposition of the correlation matrix in panel **a**. Yellow (positive) and blue (negative) loadings indicate regions of chromosome 1 partitioned into A and B compartments, respectively. **c** Gene density (genes per megabase) distributions in A (yellow) vs. B (blue) compartments genome-wide and per chromosome. Sample sizes and significance statistics provided in Supplementary Table 20. **d** Repeat classes significantly enriched by density (repeats per megabase) in A (yellow) vs. B (blue) compartments. Sample sizes and significance statistics provided in Supplementary Table 20. Each boxplot summarizes the combined (A + B) density distribution (Y-axis) per class (X axis); lower and upper bounds of each box (black) delimit the first and third quartiles, respectively, and whiskers extend to 1.5 times the interquartile range, while the median per class is represented as a filled white circle. **e** The PC3 loadings (purple line) from the repeat density matrix inversely correlate with alternating A/B-compartment loadings (green) for chromosome 1. See Supplementary Fig. 5b for all chromosomes. Purple rectangles plotted on the X axis denote subtelomeric regions, the red rectangle spans the pericentromere, and the black point marks the median centromere-associated tandem repeat position. Mb megabases. Source data are provided as a Source Data file.

consequences of genome duplication[145]. The new genome described here for *H. boettgeri*, another pipid frog, is also significant in this regard, as it enables an interesting comparison of *Xenopus* genomes to that of a closely related outgroup. Moreover, the genomes of *E. coqui* and *E. pustulosus* provide a foundation for future studies of the evolution of ontogenies and their underlying developmental mechanisms, as *E. coqui* is a direct-developing frog with no tadpole stage[16] and *E. pustulosus*, a foam-nesting frog, is a model for studying mating calls and female mate choice[18]. In addition to their interesting life histories, both frogs display distinct patterns of gastrulation[146,147]. Finally, recent work has demonstrated the efficacy of genetic or genomic analysis for understanding the impact of chytrid fungus on various amphibian species[148]. A deeper and broader understanding of amphibian genomes will be useful in the context of the global decline of amphibian populations[149,150].

*Note added in proof:* The recent finding of tetraploid dwarf clawed frogs from the Congo suggests that the diploid *Hymenochirus* we studied may distinct from *H. boettgeri*[151].

## Methods

This study complies with the ethical standards set forth by the Institutional Animal Care and Use Committee (IACUC) protocols at the University of California Berkeley, Yale University, University of Cincinnati, and the University of the Pacific. The IACUC and associated facilities are subject to review and oversight by NIH's Office of Lab Animal Welfare.

### *Xenopus tropicalis* genomic DNA extraction and sequencing

High molecular weight DNA was extracted from the blood of an $F_{17}$ *Xenopus tropicalis* Nigerian strain female[25]. Paired-end (PE) Illumina whole-genome shotgun (WGS) libraries were constructed by the QB3 Functional Genomics Laboratory (FGL) using a KAPA HyperPrep Kit and sequenced on an Illumina HiSeq 2500 as 2 × 250 bp reads by the Vincent J. Coates Genomics Sequencing Lab (VCGSL) at the University of California, Berkeley (UCB). Single-Molecule Real-Time (SMRT) continuous long-read (CLR) sequencing was performed at the HudsonAlpha Institute for Biotechnology (HAIB) on Pacific Biosciences (PacBio) RSII machines with P6-C4 chemistry (Supplementary Data 1). Chromium Genome linked-read (10x Genomics) sequencing was carried out by HAIB on an Illumina HiSeq X Ten. Hi-C libraries were constructed by Dovetail Genomics LLC. See Supplementary Note 1 for more detailed extraction and sequencing methods.

### *Xenopus tropicalis* genome assembly and annotation

Chromium linked-read (10x Genomics) data were assembled with Supernova[152] (v1.1.5). This assembly was used to seed the assembly of PacBio CLR data using DBG2OLC[153] (commit 1f7e752). An independent PacBio-only assembly was constructed with Canu[154] (v1.6-132-gf9284f8). These two assemblies were combined, or metassembled, using MUMmer[155] (v3.23) and quickmerge[156] (commit e4ea490) (Supplementary Fig. 1a). Residual haplotypic redundancy was identified and removed (Supplementary Fig. 1b). The non-redundant metassembly

was scaffolded with Sanger paired-ends and BAC-ends[45] using SSPACE[157] (v3.0) and Hi-C using 3D-DNA[117,158,159] (commit 2796c3b), then manually curated in Juicebox[160,161] (v1.9.0). The assembly was polished with Arrow[162] (smrtlink v6.0.0.47841), Pilon[163] (v1.23), and then FreeBayes[164] (v1.1.0-54-g49413aa) with ILEC (map4cns commit dd89f52, https://bitbucket.org/rokhsar-lab/map4cns). The genome was annotated with the DOE-Joint Genome Institute (JGI) Integrated Gene Call (IGC) pipeline[165] (v5.0) using transcript assemblies (TAs) generated with Trinity[166,167] (v2.5.1) from multiple developmental stages and tissues (Supplementary Data 1). RepeatModeler[168] (v1.0.11) was run on all frog species. The frog and ancestral repeat libraries from RepBase[169] (v23.12) were combined with the repeat consensuses identified by RepeatModeler. The merged repeat library was used to annotate repeats of all frogs with RepeatMasker[170] (v4.0.7). See Supplementary Note 2 for more detailed assembly and annotation methods.

### *Hymenochirus boettgeri* metaphase chromosome spread

*H. boettgeri* were obtained from Albany Aquarium (Albany, CA). Stage 26 tadpoles ($n = 10$) were incubated at room temperature in 0.01% colchicine and 1× MMR for 4–6 h. After removing the yolky ventral portion of the tadpoles, the remaining dorsal portions were pooled together in deionized water and allowed to stand for 20 min. The dorsal portions were transferred to 0.2 mL of 60% acetic acid in deionized water and allowed to stand for 5 min. The tissue was then pipetted onto a positively charged microscope slide, and excess acetic acid was blotted away. To flatten the tissue and promote chromosome spreading, the slide was covered with a coverslip, and a lead brick was placed on top of it for 5 min. The slide and coverslip were then placed on dry ice for 5 min. The coverslip was removed from the frozen slide, and the slide was stained with 0.1 mg/mL Hoechst Stain solution for 5 min. A fresh coverslip was then mounted on the slide using Vecta-Shield, and the edges were sealed with nail polish. Chromosomes in metaphase spreads (Supplementary Fig. 3a) were imaged on an Olympus BX51 Fluorescence Microscope run with Metamorph (v7.0) software using a 60× oil objective. Chromosome number was counted in 75 separate metaphase spreads.

### Genome and transcriptome sequencing of five pipanurans

Illumina PE 10x Genomics Chromium linked-read whole-genome libraries for *E. pustulosus* (from liver), *E. coqui* (from blood), and *H. boettgeri* (from liver) were sequenced on an HiSeq X at HAIB. PacBio SMRT Sequel I CLR data were generated at UC Davis DNA Technologies and Expression Analysis Core for each of *E. pustulosus* and *H. boettgeri* from liver samples. In addition, two Illumina TruSeq PE libraries (from kidney) and two Nextera mate-pair libraries (from liver) for *E. coqui* were prepared. Hi-C libraries were prepared for *H. boettgeri*, *E. pustulosus*, and *E. coqui* using the Dovetail™ Hi-C Kit for Illumina® (Beta v0.3 Short manual) following the "Animal Tissue Samples" protocol, then sequenced on a HiSeq 4000 at the VCGSL or a NextSeq at Dovetail Genomics.

Illumina TruSeq Stranded mRNA Library Prep Kit (cat# RS-122-2101 and RS-122-2102) libraries were prepared from *E. pustulosus* stages 45 and 56 whole tadpoles (gut excluded) and various adult tissues dissected from frogs maintained at the University of the Pacific. Brain ($n = 3$), dorsal skin ($n = 2$), eggs ($n = 2$), eye ($n = 2$), heart ($n = 2$), intestine ($n = 2$), larynx ($n = 3$), liver ($n = 2$), lung ($n = 2$), and ventral skin ($n = 2$) samples were washed twice with PBS, homogenized in TRIzol Reagent, and centrifuged, followed by flash freezing the supernatant. RNA was isolated following the *TRIzol Reagent User Guide* (Pub. No. MAN0001271 Rev. A.0) protocol. In addition, *H. boettgeri* eggs were homogenized in TRIzol Reagent and processed according to the manufacturer's instructions. RNA was then isolated using the QIAGEN RNeasy Mini Kit (cat# 74104). An Illumina mRNA library was prepared using the Takara PrepX RNA-Seq for Illumina Library Kit (cat# 640097) by the QB3 FGL at UCB. All libraries were sequenced at the VCGSL on an

HiSeq 4000 as $2 \times 151$ bp reads. See Supplementary Note 3 for additional details about DNA/RNA extractions and library preparations, and Supplementary Data 1 for a complete list of DNA/RNA sequencing data generated for *E. coqui*, *E. pustulosus*, and *H. boettgeri*.

### Assembly and annotation of five pipanuran genomes

*E. pustulosus* and *H. boettgeri* contigs were assembled with Supernova[152] (v2.0.1). *E. coqui* contigs were assembled with Meraculous[171,172] (v2.2.4) and residual haplotypic redundancy was removed using a custom script (align_pipeline.sh v1.0, https://github.com/abmudd/Assembly) before scaffolding with SSPACE[157] (v3.0). *E. pustulosus* and *H. boettgeri* contigs were ordered and oriented using MUMmer[155] (v3.23) alignments to PBEC-polished (map4cns commit dd89f52, https://bitbucket.org/rokhsar-lab/map4cns) DBG2OLC[153] (commit 1f7e752) hybrid contigs (Supplementary Note 3). All three assemblies were scaffolded further with linked reads and Scaff10X (v2.1, https://sourceforge.net/projects/phusion2/files/scaff10x).

*E. pustulosus* and *H. boettgeri* chromosome-scale scaffolds were constructed with Dovetail Genomics Hi-C via the HiRise scaffolder[173], followed by manual curation in Juicebox[158,160,161] v1.9.0. Due to the fragmented nature of the *E. coqui* assembly, initial chromosome-scale scaffolds were first constructed by synteny with *E. pustulosus*, then refined in Juicebox[158,160,161] v1.9.0. Gaps in the *E. pustulosus* and *H. boettgeri* assemblies bridged by PacBio reads were resized using custom scripts (pbGapLen v0.0.2, https://bitbucket.org/rokhsar-lab/xentr10/src/master/assembly) and filled with PBJelly[174] (PBSuite v15.8.24). These two assemblies were polished with FreeBayes (v1.1.0-54-g49413aa) and ILEC (map4cns commit dd89f52, https://bitbucket.org/rokhsar-lab/map4cns). A final round of gap-filling was then performed on the three assemblies using Platanus[175] (v1.2.1).

Previously published *L. ailaonicum*[30] (GCA_018994145.1) *and P. adspersus*[28] (GCA_004786255.1) assemblies were manually corrected in Juicebox[158,160,161] (v1.11.08) using their respective Hi-C and Chicago data (Supplementary Data 1). Gaps in the corrected *P. adspersus* scaffolds were resized with PacBio reads (as described above) and filled using Platanus[175] (v1.2.1) with published Illumina TruSeq PE data obtained from NCBI (PRJNA439445). As described elsewhere[176], all assemblies were screened for contaminants before scaffolding, and only final scaffolds and contigs longer than 1 kb were retained for downstream analyses. More details on assembly procedures can be found in (Supplementary Note 3).

Genomic repeats in all five species were annotated with RepeatMasker[168,170] (v4.0.7 and v4.0.9) using the repeat library generated above. Protein-coding genes were annotated for *E. coqui*, *E. pustulosus*, *H. boettgeri*, and *P. adspersus* using the DOE-JGI IGC[165] (v5.0) pipeline with homology and transcript evidence. For each respective species, newly generated RNA-seq data were combined with public *H. boettgeri*[27] (BioProject PRJNA306175) and *P. adspersus*[28] (BioProject PRJNA439445) data and *E. coqui* data (stages 7, 10, and 13 hindlimb [Harvard University]; stage 9–10 tail fin skin [French National Center for Scientific Research]). TAs used as input to IGC were assembled with Trinity[166,167] (v2.5.1) and filtered using the heuristics described in Supplementary Note 3.

### Synteny and ancestral chromosome inference

One-to-one gene ortholog set between frog proteomes was obtained from the output from OrthoVenn2[64] (https://orthovenn2.bioinfotoolkits.net) using an *E* value of $1 \times 10^{-5}$ and an inflation value of 1.5 (Supplementary Note 4). The assemblies of all frog species and axolotl were pairwise aligned against the *X. tropicalis* genome using Cactus[177] (commit e4d0859) (Supplementary Note 4). Pairwise collinear runs were merged into multiple sequence alignments with ROAST/MULTIZ[178] (v012109) in order of phylogenetic topology from TimeTree[179] (http://www.timetree.org), then sorted with LAST[180] (v979) (Supplementary Note 4).

## Phylogeny and estimation of sequence divergence

Fourfold degenerate bases of one-to-one orthologs were obtained and reformatted from the MAFFT (v7.427) alignment as described in ref. 176 (Supplementary Note 4). The maximum-likelihood phylogeny was obtained with RAxML[181] (v8.2.11) using the GTR+Gamma model of substitution with outgroup *Ambystoma mexicanum*. Divergence times were calculated with MEGA7[182] (v7.0.26) with the GTR+Gamma model of substitution using Reltime method[183].

## Chromosome evolution

A custom script[176] (cactus_filter.py v1.0, https://github.com/abmudd/Assembly) was used to extract pairwise alignments from the ROAST-merged MAF file and convert alignments into runs of collinearity. The runs of collinearity were visualized with Circos[184] (v0.69-6) (Supplementary Note 4) and JCVI[185] (jcvi.graphics.karyotype v0.8.12, https://github.com/tanghaibao/jcvi).

## Centromeres, satellites, and pericentromeric repeats

Tandem repeats were called using Tandem Repeats Finder[69] (v4.09; params: *2 5 7 80 10 50 2000 -l 6 -d -h -ngs*). To identify tandem repeats enriched in pericentromeric and subtelomeric regions, we extracted the monomer sequences of all tandem repeats overlapping the region of interest. A database of non-redundant monomers was created by making a dimer database. Dimers were clustered with BlastClust[186] v2.2.26 (*-S 75 -p F -L 0.45 -b F -W 10*). A non-redundant monomer database was created using the most common monomer size from each cluster. The non-redundant sequences were mapped to the genome with BLASTN[187] (BLAST+ v2.9.0; *-outfmt 6 -evalue 1e3*). The enriched monomeric sequences in centromeres and subtelomeres were identified by selecting the highest normalized rations of tandem sequence footprints in the region of interest over the remaining portions of the genome. For more detail, see Supplementary Note 5.

## Genetic variation

Reads were aligned with BWA-MEM[188] (v0.7.17-r1188) and alignments were processed using SAMtools[189] (v1.9-93-g0ca96a4), keeping only properly paired reads (*samtools view -f3 -F3852*) for variant calling. Variants were called with FreeBayes[164] (v1.1.0-54-g49413aa; *--standard-filters --genotype-qualities --strict-vcf --report-monomorphic*). Only biallelic SNPs with depth within mode ±1.78SDs were retained. An allele-balance filter [0.3–0.7] for heterozygous genotypes was also applied. Segmental heterozygosity/homozygosity was estimated using windows of 500 kb with 50-kb step using BEDtools[190] (v2.28.0) for pooled samples or snvrate[191] (v2.0, https://bitbucket.org/rokhsar-lab/wgs-analysis). For more detail, see Supplementary Note 2.

## GC content, gene, and repeat landscape

GC-content percentages were calculated in 1-Mb bins sliding every 50 kb. Gene densities were obtained using a window size of 250 kb sliding every 12.5 kb. The repeat density matrix for *X. tropicalis* was obtained by counting base pairs per 1 Mb (sliding every 200 kb) covered by repeat families and classes of repeats. The principal component analysis (PCA) was performed on the density matrix composed of 7253 overlapping 1-Mb bins and 3070 repeats (Supplementary Note 5). The first (PC1) and second (PC2) components were smoothed using a cubic spline method.

## Chromatin immunoprecipitation

*Xenopus tropicalis* XTN-6 cells[192] were grown in 70% calcium-free L-15 (US Biologicals cat# L2101-02-50L), pH 7.2/10% Fetal Bovine Serum/Penicillin-Streptomycin (Invitrogen cat# 15140-163) at RT. Native MNase ChIP-seq protocol was performed as described previously in Smith et al.[88]. Approximately 40 million cells were trypsinized and collected; nuclei were isolated by dounce extraction and collected with a sucrose cushion. Chromatin was digested to mononucleosomes by MNase. Nuclei were lysed and soluble nucleosomes were extracted overnight at 4 °C. Extracted mononucleosomes were precleared with Protein A dynabeads (Invitrogen cat# 100-02D) for at least 4 h at 4 °C. A sample was taken for input after pre-clearing. Protein A dynabeads were bound to 10-µg antibody (50 µg/µL final concentration of either Rb-anti-Xl Cenp-a [cross-reactive with *X. tropicalis*], Rb-anti-H4 Abcam cat# 7311, or Rb-anti-H3 Abcam cat# 1791) and incubated overnight with precleared soluble mononucleosomes at 4 °C. Dynabeads bound to 50 µg/µL final concentration of Rabbit IgG antibody (Jackson ImmunoResearch cat# 011-000-003) were collected with a magnet and washed three times with TBST (0.1% Triton X-100) before elution with 0.1% SDS in TE and proteinase K incubation at 65 °C with shaking for at least 4 h. Isolated and input mononucleosomes were size-selected using Ampure beads (Beckman cat# A63880) and prepared for sequencing using the NEBNext Ultra II DNA Library Prep Kit for Illumina (NEB cat# E7654). Three replicates were sequenced on an Illumina HiSeq 4000 lane 2 × 150 bp by the Stanford Functional Genomics Facility. PE reads were trimmed with Trimmomatic[193] (v0.39), removing universal Illumina primers and Nextera-PE indices. Processed PE reads were mapped with Minimap2[194] (v2.17-r941) against the unmasked genome reference. SAMtools[189] (v1.9-93-g0ca96a4) was used for sorting and indexing the alignments. Read counts (mapping quality [MQ] ≥ 0) per 10-kb bin (nonoverlapping) for all samples were calculated with multiBamSummary from deepTools[195] (v3.3.0). Read counts were normalized by the total number of counts in the chromosomes per sample (Supplementary Note 5). Peaks were called with MACS2[196] (v2.2.7.1) and custom scripts (https://bitbucket.org/rokhsar-lab/xentr10/src/master/chipseq).

## Recombination and extended subtelomeres

The reads from the $F_2$ mapping population[25] were aligned to the v10 genome sequence using BWA-MEM[188] (v0.7.17-r1188). Variants were called using FreeBayes[164] (v1.1.0-54-g49413aa; *--standard-filters --genotype-qualities --strict-vcf*). SNPs were filtered, and valid $F_2$ mapping sites were selected when the genotypes of the Nigerian $F_0$ and the ICB $F_0$ were fixed and different and there was a depth of at least 10 for each $F_0$ SNP. Maps were calculated using JoinMap[197] v4.1 (Supplementary Note 5, Supplementary Data 2). The variation on the linkage map was smoothed using the "not-a-knot" cubic spline function calculated every 500 kb. The Pearson correlation coefficient, $r$, was calculated between recombination rates and genomic features that include GC content, repeat densities, and densities of reported CTCF and recombination hotspots[198,199].

## Chromatin conformations and higher-order interactions

Hi-C read pairs were mapped with Juicer[158,159] (commit d3ee11b) and observed counts were extracted at 1 Mb resolution with Juicer Tools (commit d3ee11b). Centromeres were estimated manually in Juicebox[160] and refined with Centurion[200] v0.1.0-3-g985439c using ICE-balanced MQ ≥ 0 matrices (https://bitbucket.org/rokhsar-lab/xentr10/src/master/hic). Rabl-like chromatin structure was visualized with PCA from Knight–Ruiz[201]-balanced MQ ≥ 30 matrices and significance was estimated by permutation testing (10,000 iterations, one-sided $\alpha = 0.01$) using custom R[202] scripts. Rabl-like constraint between p- and q-arms was measured as the sum of square distances (SSD) in PC1-PC2 dimensions, calculated between nonoverlapping bins traveling sequentially away from the centromere. Inter-/intra-chromosomal contact enrichment analyses were quantified from MQ ≥ 30 matrices using $\chi^2$ tests in R v3.5.0 (hic-analysis.R v1.0, https://bitbucket.org/rokhsar-lab/xentr10/src/master/hic). See Supplementary Note 5 for more details.

## A/B compartments

A/B compartments were called with custom R[202] scripts (call-compartments.R v0.1.0, https://bitbucket.org/bredeson/artisanal) from Knight–Ruiz-balanced (observed/expected normalized) MQ ≥ 30 Hi-C

contact correlation matrices generated with Juicer[158,159] (Supplementary Note 5). Pearson's correlation between PC1 from the Hi-C correlation matrix and gene density was used to designate A and B compartments per chromosome.

## Reporting summary

Further information on research design is available in the Nature Portfolio Reporting Summary linked to this article.

## Data availability

Data supporting the findings of this work are available throughout the main text, Methods, Supplementary Information, Supplementary Data, or archived in Zenodo (https://doi.org/10.5281/zenodo.8393403). All newly generated assemblies, annotations, and raw data are deposited in the NCBI GenBank and SRA databases: *X. tropicalis* under BioProject accession codes PRJNA577946 and PRJNA526297, *E. coqui* under BioProject accession code PRJNA578591, *E. pustulosus* under BioProject accession code PRJNA578590, and *H. boettgeri* under BioProject accession code PRJNA578589. *L. ailaonicum* and *P. adspersus* re-assemblies were deposited at NCBI GenBank under accession DAJOPU000000000 and DYDO00000000, respectively; the versions described in this manuscript are DAJOPU010000000 [https://www.ncbi.nlm.nih.gov/nuccore/DAJOPU000000000.1] and DYDO01000000 [https://www.ncbi.nlm.nih.gov/nuccore/DYDO00000000.1]. Raw *X. tropicalis* ChIP-seq data are available at the NCBI SRA under BioProject accession code PRJNA726269 and the processed data via the NCBI GEO database under series accession GSE199671. The *E. coqui* tail fin RNA-seq data generated in this study have been deposited in the NCBI SRA database under accession code PRJNA1022815. The *E. coqui* hindlimb developmental series RNA-seq data are available under restricted access as the project is not yet published, access can be obtained by contacting Mara Laslo at ml125@wellesley.edu. Source data are provided with this paper.

## Code availability

All custom scripts used in this work are archived[203] in Zenodo at https://doi.org/10.5281/zenodo.8393403 and can be found via the project repository at https://bitbucket.org/rokhsar-lab/xentr10 (tag v1.0) or via the individual repositories linked therein: https://github.com/abmudd/Assembly, https://bitbucket.org/bredeson/artisanal, https://bitbucket.org/rokhsar-lab/map4cns, https://bitbucket.org/rokhsar-lab/wgs-analysis, https://bitbucket.org/rokhsar-lab/gbs-analysis, and https://gitlab.com/Bredeson/wombat.

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

## Acknowledgements

We thank Karen Lundy and the Functional Genomics Laboratory at the University of California Berkeley for running quality control on extracted DNA and RNA and for preparing Illumina short-insert libraries; Oanh Nguyen and the DNA Technologies and Expression Analysis Cores at the University of California Davis Genome Center for preparing and sequencing PacBio libraries; Dovetail Genomics for providing the Hi-C library preparation kit, running quality control on Hi-C libraries, and preparing and sequencing Hi-C libraries; Shana McDevitt and the Vincent J. Coates Genomics Sequencing Laboratory at the University of California Berkeley for sequencing Hi-C and Illumina short-insert libraries; Shengqiang Shu for advice on the use of the IGC annotation pipeline. We thank Rick Elinson for providing *E. coqui* frogs and tissues. We thank Gary Gorbsky from the Oklahoma Medical Research Foundation and Marko Horb and the National *Xenopus* Resource at the MBL for providing the XTN-6 cell lines. We also thank Chunhui Hou and colleagues for permission to access their Hi-C data before publication. This study was supported by NIH grants R01HD080708 to D.S.R.; R01GM086321, R01HD065705 to D.S.R. and R.M.H.; R35GM127069 to R.M.H.; R35 GM118183 to R.H. A.B.M. was supported by NIH grants T32GM007127 and T32HG000047 and a David L. Boren Fellowship. D.S.R. is grateful for support from the Marthella Foskett Brown Chair in Biological Sciences; R.M.H., the C.H. Li

Distinguished Chair in Molecular and Cell Biology; and R.H., the Flora Lamson Hewlett chair in biochemistry. A.F.S. and O.K.S. were supported by R01GM074728, O.K.S. by NIH T32 GM113854-02 and NSF GRFP; M.K.K. and M.Lane by R01HD102186; J.H. by NSF grants DEB-1701591 and DBI-1702263; M.Laslo, a Graduate Women in Science Fellowship; T.K. by the Basic Science Research Program, National Research Foundation of Korea (NRF), Ministry of Education (2018R1A6A1A03025810), Future-leading Project Research Fund (1.200094.01) of UNIST and the Institute for Basic Science (IBS-R022-D1); J.B.W. and H.S.P. by R01GM104853, R01HD085901; M.J.R. by NSF IOS-0910112; Smithsonian Tropical Research Institute; Clark Hubbs Regents Professorship; L.M.S. by the "Centre National de la Recherche Scientifique" (PEPS ExoMod "Triton") and the "Muséum National d'Histoire Naturelle" (Action Transversale du Muséum "Cycles biologiques: Evolution et adaptation") and a Scientific council post-doctoral position to G.K. This work used the Vincent J. Coates Genomics Sequencing Laboratory at the University of California Berkeley, supported by NIH grant S10OD018174, and the DNA Technologies and Expression Analysis Cores at the University of California Davis Genome Center, supported by NIH grant S10OD010786. This research used the National Energy Research Scientific Computing Center, a Department of Energy Office of Science User Facility supported by contract number DE-AC02-05CH11231. L.M.S. acknowledges the "Ecole Normale Supérieure de PARIS" genomic platform for RNA sequencing and the PCIA high-performance computing platform at "Muséum National d'Histoire Naturelle".

## Author contributions

J.V.B., A.B.M., S.M.R., T.M., R.M.H. and D.S.R. wrote the manuscript with feedback from M.Laslo, H.P.S., J.H., J.B.L., J.B.W., M.J.R., O.K.S., D.R.B., M.G.P., J.H., N.B., T.K., L.M.S., R.H., J.S., M.K.K., A.F.S. and D.H. Genomes were assembled by J.V.B., S.S.B. (Xtr); A.B.M., and K.C.B. (other frogs). S.M.R., A.B.M. and G.K. assembled transcripts and annotated genomes. S.M.R. and J.V.B. assessed gene completeness; S.M.R. analyzed repeat and recombination landscapes. S.M.R. and J.P. identified centromeric repeats. O.K.S., G.A.F. and A.F.S. conducted ChIP-seq experiments, and S.M.R. performed analysis. J.V.B. analyzed Hi-C features. T.M. constructed the linkage map. T.M. and J.V.B. analyzed heterozygosity. A.B.M. performed genome-wide comparisons. K.E.M. and R.H. examined Hbo metaphase spreads. M.K.K. and M.Lane inbred Xtr frogs. R.M.H. (Xtr); M.G.P. (Epu); K.E.M. and R.H. (Hbo); M.Laslo and J.H. (Eco) collected frogs. R.M.H. (Xtr); M.G.P., H.S.P. (Epu); and D.R.B. (Eco) collected tissue samples. A.B.M., D.R.B. (Eco); J.B.L. and I.P. (Xtr) extracted DNA. A.B.M., S.M.R. (Epu); K.E.M., R.H. (Hbo); and L.M.S. (Eco) extracted RNA and libraries were prepared by A.B.M. (Epu). M.Laslo, J.H. (Eco); K.E.M. and R.H. (Hbo) provided RNA-seq data. T.K., M.J.R., J.B.W. (Epu); and J.B.L. (Xtr) coordinated sequencing. C.P., J.G. and J.S. prepared and sequenced 10x Genomics, PacBio, and Illumina mate-pair libraries. D.H. prepared Hi-C libraries. R.D.D. and J.H.M. provided early access to the Pad assembly. N.B. (Eco) provided bioinformatic support. L.M.S. led the Eco efforts. R.M.H. and D.S.R. led the project.

## Competing interests

D.S.R. is a member of the Scientific Advisory Board of, and a minor shareholder in, Dovetail Genomics LLC, which provides as a service the high-throughput chromatin conformation capture (Hi-C) technology used in this study. M.K.K. is President and co-founder of Victory Genomics, Inc. The remaining authors declare no competing interests.

## Additional information

¹Department of Molecular and Cell Biology, Weill Hall, University of California, Berkeley, CA 94720, USA. ²DOE-Joint Genome Institute, 1 Cyclotron Road, Berkeley, CA 94720, USA. ³Department of Biochemistry, Stanford University School of Medicine, 279 Campus Drive, Beckman Center 409, Stanford, CA 94305-5307, USA. ⁴Computer Science Division, University of California Berkeley, 2626 Hearst Avenue, Berkeley, CA 94720, USA. ⁵HudsonAlpha Genome Sequencing Center, HudsonAlpha Institute for Biotechnology, Huntsville, AL 35806, USA. ⁶Pediatric Genomics Discovery Program, Departments of Pediatrics and Genetics, Yale University School of Medicine, 333 Cedar Street, New Haven, CT 06510, USA. ⁷Department of Organismic and Evolutionary Biology, and Museum of Comparative Zoology, Harvard University, Cambridge, MA 02138, USA. ⁸Département Adaptation du Vivant, UMR 7221 CNRS, Muséum National d'Histoire Naturelle, Paris, France. ⁹Department of Biological Sciences, University of Cincinnati, Cincinnati, OH, USA. ¹⁰Department of Biomedical Engineering, Ulsan National Institute of Science and Technology, Ulsan 44919, Republic of Korea. ¹¹Center for Genomic Integrity, Institute for Basic Science (IBS), Ulsan 44919, Republic of Korea. ¹²Department of Integrative Biology, Patterson Labs, 2401 Speedway, University of Texas, Austin, TX 78712, USA. ¹³Department of Biological Sciences, University of the Pacific, 3601 Pacific Avenue, Stockton, CA 95211, USA. ¹⁴Department of Molecular and Cell Biology and Institute of Systems Genomics, University of Connecticut, 181 Auditorium Road, Unit 3197, Storrs, CT 06269, USA. ¹⁵Department of Molecular Biosciences, Patterson Labs, 2401 Speedway, The University of Texas at Austin, Austin, TX 78712, USA. ¹⁶Innovative Genomics Institute, University of California, Berkeley, CA 94720, USA. ¹⁷Chan-Zuckerberg BioHub, 499 Illinois Street, San Francisco, CA 94158, USA. ¹⁸Okinawa Institute of Science and Technology Graduate University, Onna, Okinawa 9040495, Japan. ¹⁹These authors contributed equally: Jessen V. Bredeson, Austin B. Mudd, Sofia Medina-Ruiz. ✉e-mail: dsrokhsar@gmail.com

