## [Peer Review File · Nature Communications]

Conserved chromatin and repetitive patterns reveal slow genome evolution in frogsREVIEWER COMMENTS

Reviewer #1 (Remarks to the Author):

The manuscript by Bredeson, Mudd, Medina-Ruiz et al., (“Conserved chromatin and repetitive patterns reveal slow genome evolution in frogs”) provides a more complete chromosome-scale assembly of the *Xenopus tropicalis* genome, as well as draft assemblies for the genomes of 3 other frogs, *Hymenochirus boettgeri*, *Engystomops pustulosus*, and *Eleutherodactylus coqui*. Comparisons of these assemblies reveal distinct aspects of anuran chromosomal evolution, including an inferred ancestral chromosome number, conservation of synteny with mammalian genomes, accumulation of extensive subtelomeric repetitive regions, and a marked stability of chromosomal organization relative to mammals and birds. This is a substantive contribution to our understanding of vertebrate genome evolution, and it will draw attention across a spectrum of evolutionary biology and genomics, genetics, and cell biology, among other areas.

These studies are very well done, using a range of well-documented computational and experimental approaches. Integration of data from multiple sequencing strategies has yielded a major advance in the quality of the *X. tropicalis* genome assembly, resolving several long-standing issues.

The manuscript and supplementary material incorporate a massive amount of data, including genomic sequence data, karyotype, and Hi-C analyses, and thus it would seem unreasonable to request additional studies. While the manuscript is focused on specific aspects of chromosomal evolution, however, several topics of considerable interest are not discussed, particularly comparisons of gene sequences (ORFs) among different species. While it may be premature to make such comparisons, given the early state of the leptodactylid draft genomes, many would be quite interested to know, e.g., which sets of genes show the greatest sequence divergence, or whether there are features of coding genes or genome organization that correspond to alterations in life history. A limited summary of such features, with a promise of a subsequent detailed analysis, would be a welcome addition (see Kolora, Owens, et al., 2021; comparison of *Sebastes* genomes).

The paper overall would be strengthened by more comparative information, in terms of where this clade is consistent with, and where it differs from, other vertebrate taxa. For example, it is unclear whether the hi-C findings reveal something exceptional about pipanuran chromosomal organization, or whether these results are consistent with observations in other vertebrates. Some additional comparative context would be useful here. In addition, there are a number of points that call for clarification (see below).

Specific questions:

Does the repeat structure / terminal region contribute to chromosome stabilization over evolutionary time?

In comparisons of anuran, avian, and mammalian genomes, what structural features are associated with variation in chromosome number, or the emergence of minichromosomes?

P6 (lines 103-105): With regard to sequence that was “missed” in previous attempts, are there structural characteristics of sequence that had previously escaped analysis (repeat content, spatial distribution, etc) that could explain why it had been missed? (is this likely to be an issue for vertebrate genome assemblies generally?)

With regard to evolutionary changes in centromeres: Have these comparisons yielded insight into how non-dominant centromeres become inactivated following translocation? Are there structural differences between mature/conserved centromeres and evolutionarily “younger” centromeres? (p13, lines 274-276): Are there any unique characteristics of centromere D that might contribute to persistence (or resistance to silencing)?

P14, line 300; With regard to regions showing little to no recombination, the initial SSLP map of *X. tropicalis* (Wells et al., 2011) showed no evidence of recombination on the P arm of Chr. 2, and considered a selection-based argument for the restriction of recombination in this region. Is this consistent with the current findings?

What are the consequences for a high rate of recombination within the subtelomeric regions? Would this be likely to contribute to expansion? This issue is discussed, but this specific point wasn't clear.

P 19, Lines 422-427: How common are interchromosomal contacts in mammalian cells? Again, this would be easier to interpret with more comparative information.

P 19: With regard to the increased constraint on embryonic chromosomes in a Rab1 configuration result from the temporal limitations of the cell cycle? Are these constraints typical of actively cycling cells?

P 20, line 467: What is the average size of A vs B compartments in *X. tropicalis*?

P 21, lines 499-504: Do the domains enriched for chromosomal contacts correspond to regions of increased recombination?

Reviewer #2 (Remarks to the Author):

In this really interesting MS, the authors present genomic sequencing of a set of frogs, using HiC data to deliver good, chromosomally-complete assemblies with very high contig N50 metrics: a tour de force. The subsequent analyses are well formed and ask and in large measure answer a series of biologically important questions.

It is striking that large-scale synteny (or linkage) has been maintained in this group, and that gene order (or local synteny) has also been constrained from change. This in itself is an important discovery. It has been proposed previously from genetic mapping data, and comparisons of draft genomes scaffolded by genetic maps [1], but to see it laid out so clearly in the assemblies is very strong.

The patterns of fusion and breakage are very well described, but given the limited change observed (and likely observable) it is harder to derive "significance" from the resulting patterns, and thus to define meaningful "process". I wonder what a set of frog karyotypes might look like if evolved under an entirely neutral/stochastic model of simple low rates of fusion and fission (random across chromosomes).

One consequence of the fission and fusion process is the rate of crossovers per DNA base (cM per Mb). There are many shifts in chromosome (arm) size on the tree: do these shifts correspond to changed evolutionary rates for the loci affected (as would be predicted if recombination is one of the major drivers of mutation)? Element H is one of several that differ between the two major clades in the tree...

The Rabl like patterning of the interchromosomal HiC contacts could usefully be contextualised by the work of the Dudchenko/Erez Lieberman/Rowland labs in exploring patterns across Metazoa [2]: where do frogs fit into their models (neatly, it would seem).

I would have liked to see more explicit discussion of the runs of homozygosity noted in the Xt genome

1 see <https://academic.oup.com/g3journal/article/11/12/jkab286/6353606?login=true> [which includes coauthors from the current study I see]

2 see <https://www.science.org/doi/10.1126/science.abe2218>

Smaller comments.

Abstract

"phylogenetically ancient group of living amphibians"

- an odd phrasing; I see what is meant; Perhaps "Extant frogs"?

Abstract and introduction; also elsewhere

"emerging model frog species"; "the túngara frog (*Engystomops pustulosus*), which is a model for vocalization, and the Zaire dwarf clawed frog (*Hymenochirus boettgeri*), which has an unusually small embryo and is a model for regulation of cell and body sizes."; "three new emerging model species"[results]

- if it is claimed these are "models" I would expect to see a body of work referenced. I think the authors mean "would be an interesting system in which one could study...". Under this definition of "model" all species are models for their unique or shared biology). I suggest remove the assertion of model status if there is no work supporting this.

"occurred along frog phylogeny"

- during phylogeny

"shotgun cloning and/or sequences"

- and/or sequencing

"gaps in the earlier genome assemblies arising from cloning biases in the Sanger sequencing process" p8

- likely arising (this is an assertion)

Figure 1

It would be informative to have (perhaps as a panel in Fig1) a tree of anurans illustrating the taxonomic relationships of the various groups mentioned in the results section (Pipoidea, Neobatrachia, Leiopelmatoidea, etc etc). This tree could also include the relative inferred timing of the splits discussed in the text.

Karyotypic stability... perhaps needs more discussion - is this a taxonomically restricted trait?

Pipoidea

[https://goat.genomehubs.org/search?query=tax_tree\(30319\) AND haploid_number AND tax_rank\(species\)&result=taxon&fields=haploid_number&includeEstimates=false&summaryValues=count&taxonomy=ncbi&report=histogram&rank=species#tax_tree\(30319\) AND haploid_number AND tax_rank\(species\)](https://goat.genomehubs.org/search?query=tax_tree(30319) AND haploid_number AND tax_rank(species)&result=taxon&fields=haploid_number&includeEstimates=false&summaryValues=count&taxonomy=ncbi&report=histogram&rank=species#tax_tree(30319) AND haploid_number AND tax_rank(species))

Neobatrachia

[https://goat.genomehubs.org/search?query=tax_tree\(Neobatrachia\) AND haploid_number AND tax_rank\(species\)&result=taxon&fields=haploid_number&includeEstimates=false&summaryValues=count&taxonomy=ncbi&report=histogram&rank=species#tax_tree\(Neobatrachia\) AND haploid_number AND tax_rank\(species\)](https://goat.genomehubs.org/search?query=tax_tree(Neobatrachia) AND haploid_number AND tax_rank(species)&result=taxon&fields=haploid_number&includeEstimates=false&summaryValues=count&taxonomy=ncbi&report=histogram&rank=species#tax_tree(Neobatrachia) AND haploid_number AND tax_rank(species))

"We note that the D centromere persists in both end-to-end fusions involving D, suggesting that centromeres derived from different ancestral elements may be differentially susceptible to silencing."

- n=2 and thus this "observation" is not really worth mentioning...

Figure 2

"Chromosomes are centered by the position of centromeric tandem repeats (black dot and dotted vertical line)."

- I dont see the dot and line

"the X. tropicalis centromeric satellite repeat from tandem repeat analysis with a red star"

- not visible in chr7?

Heterozygosity

"15 long heterozygous blocks ranging in size from 1.34 to 74.6 Mb."

- the relevance of the sequencing of geographic populations is not made clear in the main text; are these retained Hz blocks derived specifically from one population or the other? Are they evident in the parental populations? How "related" are the presumed parental populations to the individuals set up to start the inbreeding? Is this more or less than one would expect from the inbreeding (this is modellable)?

REVIEWER COMMENTS

Reviewer #1 (Remarks to the Author):

The manuscript by Bredeson, Mudd, Medina-Ruiz et al., (“Conserved chromatin and repetitive patterns reveal slow genome evolution in frogs”) provides a more complete chromosome-scale assembly of the *Xenopus tropicalis* genome, as well as draft assemblies for the genomes of 3 other frogs, *Hymenochirus boettgeri*, *Engystomops pustulosus*, and *Eleutherodactylus coqui*. Comparisons of these assemblies reveal distinct aspects of anuran chromosomal evolution, including an inferred ancestral chromosome number, conservation of synteny with mammalian genomes, accumulation of extensive subtelomeric repetitive regions, and a marked stability of chromosomal organization relative to mammals and birds. This is a substantive contribution to our understanding of vertebrate genome evolution, and it will draw attention across a spectrum of evolutionary biology and genomics, genetics, and cell biology, among other areas.

These studies are very well done, using a range of well-documented computational and experimental approaches. Integration of data from multiple sequencing strategies has yielded a major advance in the quality of the *X. tropicalis* genome assembly, resolving several long-standing issues.

Thanks!

The manuscript and supplementary material incorporate a massive amount of data, including genomic sequence data, karyotype, and Hi-C analyses, and thus it would seem unreasonable to request additional studies. While the manuscript is focused on specific aspects of chromosomal evolution, however, several topics of considerable interest are not discussed, particularly comparisons of gene sequences (ORFs) among different species. While it may be premature to make such comparisons, given the early state of the leptodactylid draft genomes, many would be quite interested to know, e.g., which sets of genes show the greatest sequence divergence, or whether there are features of coding genes or genome organization that correspond to alterations in life history. A limited summary of such features, with a promise of a subsequent detailed analysis, would be a welcome addition (see Kolora, Owens, et al., 2021; comparison of *Sebastes* genomes).

While these are very interesting questions, they are beyond the scope of the present study, which focused on chromosome structure, organization, recombination, and evolution. In particular, it would be difficult to provide a limited summary of gene-based evolutionary features without extensive additional analysis.

The paper overall would be strengthened by more comparative information, in terms of where this clade is consistent with, and where it differs from, other vertebrate taxa. For example, it is unclear whether the hi-C findings reveal something exceptional about pipanuran chromosomal organization, or whether these results are consistent with observations in other vertebrates. Some additional comparative context would be useful here.

We have expanded our previously brief discussions to compare our results on chromatin organization, recombination, and karyotype stability to findings in other groups, notably birds and mammals where the most comparative studies have been done. In summary:

1. **Chromatin organization.** Taken together, these intra- and inter-chromosome contacts are consistent with a Rabl-like (Type-I¹) chromosome configuration^{2,3} in *Xenopus* blood cells. Outside of mammals, Rabl-like contacts have been observed in a wide diversity of taxa. We note, however, that Hi-C patterns can depend on cell type, cell cycle stage, and developmental time; while Rabl-like HiC patterns are often absent from tissue samples used in mammalian genome sequencing projects, they have been observed in studies of mouse and human cell lines. We have expanded this discussion in the main text (page 22) and added a short note to **Supplementary Note 5** (pages 65–66):

Hoencamp et al.¹ surveyed 24 plant and animal species using Hi-C and observed Rabl-like patterns in 14 (58.3%) of them: *Xenopus laevis*, *Ciona robusta*, (formerly *C. intestinalis*), *Aedes aegypti*, *Culex quinquefasciatus*, *Drosophila melanogaster*, *Hypsibius dujardini*, *Clonorchis sinensis*, *Cristatella mucedo*, *Acropora millepora*, *Pleurobrachia bachei*, *Agaricus bisporus*, *Saccharomyces cerevisiae*, *Arachis hypogaea*, *Triticum aestivum* (see <https://tiny.3dg.io/hdeb2021s2>). Out of 7 vertebrates sampled in this study, however, only *Xenopus laevis* fibroblasts showed a Rabl-like pattern. Rabl-like Hi-C patterns have also been observed in single-cell organisms, such as baker's yeast⁴ (as noted above) and apicomplexan parasites⁵. Such a broad sampling of species with Rabl-like Hi-C patterns suggests that this chromosome configuration may be a primitive feature of eukaryotic chromosome biology.

In addition to Hoencamp et al., numerous mammalian genome analyses conspicuously lack such Rabl-like patterns, including muntjak⁶ fibroblasts; eight human cell types⁷; Rhesus macaque⁸ female fibroblasts; mouse CH12-LX B-Lymphoblasts⁷, Patski cell lines⁸, and activated B cells⁹; California sea lion (*Zalophus californianus*) whole blood^{10,11}; Cheetah (*Acinonyx jubatus*) blood^{11,12}; Collared lemur (*Eulemur collaris*) liver tissue¹¹; Red panda (*Ailurus fulgens*) blood^{11,13}; Virginia opossum (*Didelphis virginiana*) blood¹¹; and Red kangaroo (*Macropus rufus*) blood¹¹. The presence of Rabl-like Hi-C patterns, however, evidently depends on cell type and cell cycle stage with Rabl-like patterns observed in mouse hematopoietic cells¹⁴, haploid G1-phase ES cells¹⁵, diploid embryonic stem cells¹⁵, and peripheral blood mononuclear cells¹⁵.

2. **Recombination.** We find here that recombination in *X. tropicalis* is (1) heavily concentrated at the ends of chromosomes and (2) is nearly absent in the central regions of chromosomes, including but extending far beyond the immediate vicinity of the centromere. This pattern appears to be common in the macro-chromosomes of birds (chicken and zebra finch), the longer chromosomes of mammals (human, pig, dog), and in snakes, and so may be a common feature of vertebrate macrochromosomes, regardless of the use of PRDM9 in initiating recombination. The generality of this pattern

does not seem to have been noted previously. We have rewritten and expanded this discussion in the main text (page 19) to emphasize this general pattern.

- 3. Karyotype stability.** We have added a brief paragraph to the main text (page 16) noting that chromosomal conserved synteny across pipanuran frogs is comparable to that observed in birds, which have evolved by limited intra-chromosomal rearrangement from an $n = 40$ ancestor¹⁶, mostly involving fusion of microchromosomes, as we find here for pipanurans (see below). The relative stasis of frog and bird chromosomes is in contrast to the variable karyotypes of mammals, which was first noted by Bush et al.¹⁷ and is now extensively documented at the level of chromosomal painting¹⁸ and genome sequence¹⁹. The reasons for these different modes of evolution remain unclear but are likely related to the difficulty in fixing partial-arm chromosomal rearrangements in large historically panmictic populations due to reduced fertility in translocation heterozygotes, as first noted by Wright²⁰. Partial-arm rearrangements, as observed in mammals, can become fixed in populations that are dynamically subdivided by local extinction and colonization, which allows the reduced fertility of translocation heterozygotes to be overcome by genetic drift²¹.

In addition, there are a number of points that call for clarification (see below).

Specific questions:

Does the repeat structure / terminal region contribute to chromosome stabilization over evolutionary time?

While this is an interesting suggestion, it is not clear how we could test it with the data in hand. We report that the phenomenon of extended repetitive distal arms and chromosome stability co-occur, but any claim about causation would be speculative. Alternately, it could go the other way: the long-term stability of the chromosomes, and the enrichment of subtelomeres for recombination, could promote the expansion of tandemly repeated elements.

In comparisons of anuran, avian, and mammalian genomes, what structural features are associated with variation in chromosome number, or the emergence of minichromosomes?

This is an interesting question. We and others have previously shown that microchromosomes are homologous across gnathostomes^{22–24} and—after accounting for the early vertebrate genome duplications—also conserved in lamprey, and even amphioxus²³. Small chromosomes are therefore ancestral to jawed vertebrates, as has been previously suggested²³.

P6 (lines 103-105): With regard to sequence that was “missed” in previous attempts, are there structural characteristics of sequence that had previously escaped analysis (repeat content, spatial distribution, etc) that could explain why it had been missed? (is this likely to be an issue for vertebrate genome assemblies generally?)

We attempted to address this issue briefly in the original submission and have slightly modified the revised main text (page 6) to make our findings more explicit. As shown in revised **Supplementary Fig. 2**, genomic regions that were missed by Sanger sequencing are concentrated in the extended subtelomeres (see the “Lack Sanger” track immediately below “GC%”) and are often overlapped/surrounded by blocks of tandem repeats (**Supplementary Fig. 2d–f**). **Supplementary Note 2** notes that of the 140.5k regions uncovered by Sanger sequences (mean size 486 bp), 83.4% overlap repetitive regions, and 58.3% of these repeats correspond to satellite repeats. The legend of **Supplementary Fig. 2** notes that of the 4,718 CDS (6,774 CDS exons) overlapped with regions without Sanger data, 3,511 are within 2kb of tandem repeats.

This effect of Sanger sequencing seems likely to be of historical interest since long-read single molecule sequencing like PacBio does not have the same biases as cloning-based Sanger sequencing.

With regard to evolutionary changes in centromeres: Have these comparisons yielded insight into how non-dominant centromeres become inactivated following translocation? Are there structural differences between mature/conserved centromeres and evolutionarily “younger” centromeres? (p13, lines 274-276): Are there any unique characteristics of centromere D that might contribute to persistence (or resistance to silencing)?

The evolution of differences between centromere ‘strength’ is an interesting question for future study, ideally with high-quality (gap-free) centromeric genome assemblies and extensive supporting experimental data that is beyond the scope of the present work.

We note, however, that we find that the extant centromeres of all the frog species we studied have been inherited from the pipanuran ancestor, and in that sense are equally ancient – none are “younger” than others. We do find that some ancient centromeres have been lost following end-to-end fusions, but this is expected based on the instability of dicentric chromosomes. We find that some relicts of inactivated centromeres retain evidence of pericentromere-associated LINE elements, but this does not give us insight into the mechanism of inactivation.

Regarding centromere D, we find that it is the centromere that persists in the two cases where ancestral chromosome D has fused to another chromosome. As pointed out by another reviewer, however, this is an $n = 2$ statement, so that the apparent persistence or resistance-to-silencing of the D centromere is not statistically significant based on our data. We now make this clear (main text, page 14) but continue to call out the persistence of D as a curiosity.

P14, line 300; With regard to regions showing little to no recombination, the initial SSLP map of *X. tropicalis* (Wells et al., 2011) showed no evidence of recombination on the P arm of Chr. 2, and considered a selection-based argument for the restriction of recombination in this region. Is

this consistent with the current findings?

Thanks for this question. We have added a short note to **Supplementary Note 5** (pages 60–61) about the difference between the levels of recombination that we found on 2p vs the complete absence of this arm from the earlier map described in Wells et al.²⁵. Briefly, Wells et al. noted that while they found scaffolds in the early draft genome that they could cytogenetically map to 2p (reported by Macha et al.²⁶), nearly three-quarters of genotyped SSLP markers on these scaffolds were non-polymorphic in their F2 mapping panel. Wells et al. compared these scaffolds to size-mapped controls and concluded that “These findings suggest that the level of polymorphism on the p arm of Chr. 2 [in their mapping population] is substantially reduced relative to the regions represented on the map” (note in square brackets added).

Such a situation could occur if the two parents of the F₂ population used by Wells et al. were cryptically related and shared haplotypes on chromosome 2p. Thus the discrepancy between our findings and theirs could have a simple explanation in terms of the difference between the founders of our respective mapping populations. Previously, we²⁷ inferred the existence of a recessive lethal allele in the ICB population on chromosome 2 that could lead to distorted segregation. It is possible that these phenomena are related, but fully deciphering the absence of 2p in the Wells et al.²⁵ map would require a more comprehensive survey of variation in their *X. tropicalis* cross that is beyond the scope of our study.

What are the consequences for a high rate of recombination within the subtelomeric regions? Would this be likely to contribute to expansion? This issue is discussed, but this specific point wasn't clear.

We have added the following paragraph to the main text (page 21) to clarify this point. “We hypothesize that the high rate of recombination in the extended subtelomeres of frog chromosomes drives tandem repeat expansion through illegitimate homologous recombination and, in the process, increases GC content (**Supplementary Fig. 14d,e**). Unfortunately, it is difficult to resolve cause and effect with observational data, and we cannot rule out the alternative hypothesis that short sequence motifs (**Supplementary Fig. 12b**) promote DNA breakage and then be repaired by homologous recombination.”

P 19, Lines 422-427: How common are interchromosomal contacts in mammalian cells? Again, this would be easier to interpret with more comparative information.

Thanks for this suggestion. Although widespread among diverse eukaryotes, among the 7 vertebrates studied by Hoencamp, only *X. laevis* fibroblasts show Rab1-like patterns. Rab1-like patterns have been described in mammals, but only in some cell types/under some conditions, and are more typically absent. We have added a new section under **Supplementary Note 5** (pages 65–66) that summarizes this extensive literature. This Note is copied below for your convenience:

Hoencamp et al.¹ surveyed 24 plant and animal species using Hi-C and observed Rabl-like patterns in 14 (58.3%) of them: *Xenopus laevis*, *Ciona robusta*, (formerly *C. intestinalis*), *Aedes aegypti*, *Culex quinquefasciatus*, *Drosophila melanogaster*, *Hypsibius dujardini*, *Clonorchis sinensis*, *Cristatella mucedo*, *Acropora millepora*, *Pleurobrachia bachei*, *Agaricus bisporus*, *Saccharomyces cerevisiae*, *Arachis hypogaea*, *Triticum aestivum* (see <https://tiny.3dg.io/hdeb2021s2>). Out of 7 vertebrates sampled in this study, however, only *Xenopus laevis* fibroblasts showed a Rabl-like pattern. Rabl-like Hi-C patterns have even been observed in single-cell organisms, such as baker's yeast⁴ and apicomplexan parasites⁵. Such a broad sampling of species with Rabl-like Hi-C patterns suggests that this chromosome configuration may be a primitive feature of eukaryotic chromosome biology.

In addition to Hoencamp et al., numerous mammalian genome analyses conspicuously lack such Rabl-like patterns, including muntjak⁶ fibroblasts; eight human cell types⁷; Rhesus macaque⁸ female fibroblasts; mouse CH12-LX B-Lymphoblasts⁷, Patski cell lines⁸, and activated B cells⁹; California sea lion (*Zalophus californianus*) whole blood^{10,11}; Cheetah (*Acinonyx jubatus*) blood^{11,12}; Collared lemur (*Eulemur collaris*) liver tissue¹¹; Red panda (*Ailurus fulgens*) blood^{11,13}; Virginia opossum (*Didelphis virginiana*) blood¹¹; and Red kangaroo (*Macropus rufus*) blood¹¹. The presence of Rabl-like Hi-C patterns, however, evidently depends on cell type and cell cycle stage with Rabl-like patterns observed in mouse hematopoietic cells¹⁴, haploid G1-phase ES cells¹⁵, diploid embryonic stem cells¹⁵, and peripheral blood mononuclear cells¹⁵.

P 19: With regard to the increased constraint on embryonic chromosomes in a Rabl configuration result from the temporal limitations of the cell cycle? Are these constraints typical of actively cycling cells?

These are great questions. Indeed, we believe that the stronger Rabl-like patterns we observe in the embryonic stage datasets are the result of the rapidly-dividing nature of these cells. Dernburg and colleagues²⁸ reasoned that the Rabl configuration observed in *Drosophila* embryonic nuclei^{29,30} is a result of anaphase chromosome movement and, due to their rapidly-dividing nature, such chromosomes are unable to “relax” into a diffused chromatin state. These patterns are even observed in early mouse embryonic cells^{31,32}. This Rabl structure may not be universal, however, as one study³³ in *Drosophila* observed that intrachromosomal interarm contacts in embryo-derived Kc167 cells were no more enriched over background in contrast to those previously reported in other embryonic nuclei. We have expanded the discussion in the main text (pages 22–24) with the above points.

(Please note that, throughout the main text, we have converted all uses of “Rabl” to “Rabl-like,” as we cannot show with the current data—despite evidence of centromere-centromere and telomere-telomere clustering—that the centromeres and telomeres are situated away-from and nearer-to the nucleolus, respectively, characteristic of the *sensu stricto* Rabl

configuration³.)

P 20, line 467: What is the average size of A vs B compartments in *X. tropicalis*?

Despite their two-fold difference in gene content, A and B compartment lengths are comparable, with approximately exponential length distributions that are now shown in **Supplementary Fig. 17**. The arithmetic mean sizes are A = 1.32 Mb, B = 1.48 Mb; the corresponding geometric means (i.e., the exponential of the arithmetic mean of logarithms of segment lengths) are somewhat shorter (A = 0.807 Mb, B = 0.946 Mb). This statement has been added to the main text (page 25).

P 21, lines 499-504: Do the domains enriched for chromosomal contacts correspond to regions of increased recombination?

As noted in lines 377–387 of the original submission, we find an approximately >10-fold enrichment in recombination in the subtelomeres relative to the remaining chromosomal length. We have added in **Supplementary Note 5** (pages 60–61) a discussion of the statistical significance of this enrichment ($p < 3 \times 10^{-317}$, using a Kolmogorov-Smirnov test with recombination rate) and included panel to show recombination rate distributions per chromosome domains (**Supplementary Fig. 10c**). In lines 499–504 of the original submission, as noted by the reviewer, we find a relatively weak enrichment for p-p and q-q inter-chromosomal contacts, but with our data, we do not have the resolution to localize these inter-chromosomal contact enrichments specifically to the recombinogenic regions.

Reviewer #2 (Remarks to the Author):

In this really interesting MS, the authors present genomic sequencing of a set of frogs, using Hi-C data to deliver good, chromosomally-complete assemblies with very high contig N50 metrics: a tour de force. The subsequent analyses are well formed and ask and in large measure answer a series of biologically important questions.

It is striking that large-scale synteny (or linkage) has been maintained in this group, and that gene order (or local synteny) has also been constrained from change. This in itself is an important discovery. It has been proposed previously from genetic mapping data, and comparisons of draft genomes scaffolded by genetic maps [1], but to see it laid out so clearly in the assemblies is very strong.

Thanks, we also find it remarkable! We have added a citation in the introduction (main text, page 4) to the map-based demonstration of chromosome-scale conserved synteny between *Bombina variegata* and *Xenopus tropicalis*³⁴:

Nürnberg, B. *et al.* A dense linkage map for a large repetitive genome: discovery of the sex-determining region in hybridizing fire-bellied toads (*Bombina bombina* and *Bombina variegata*). *G3* **11**, (2021).

We have also added a brief discussion to the main text (page 12) comparing the *Bombina* findings with our genome sequence-based analysis. *Bombina* belongs to the sister group to pipanuran, and implies that many of our findings have a broader phylogenetic footprint. Specifically (as explained below) we find that nine of the twelve *B. variegata* chromosomes correspond directly to nine of our thirteen ancestral pipanuran units, which implies that these nine were also ancestral chromosomes (or chromosome arms) of the *Bombina*+pipanuran clade (which does not have a common name). The other three *B. variegata* chromosomes have a more complex relationship with the remaining four ancestral pipanuran units; notably, BVA1 comprises portions of multiple pipanuran ancestral units.

- BVA2 \equiv XTR1 \equiv ancestral pipanuran chromosome A
- BVA3 \equiv XTR2 \equiv ancestral pipanuran chromosome B
- BVA4 \equiv XTR3 \equiv ancestral pipanuran chromosome C
- BVA5 \equiv XTR5 \equiv ancestral pipanuran chromosome F
- BVA6 \equiv XTR6 \equiv ancestral pipanuran chromosome G

Nürnberg et al. found that XTR7 is syntenically equivalent to a fusion of BVA9 and BVA8. We find that the metacentric XTR7 is a centric (i.e., Roberstonian) fusion of ancestral pipanuran units I and H. It follows that:

- BVA9 \equiv XTR7p \equiv ancestral pipanuran chromosome I
- BVA8 \equiv XTR7q \equiv ancestral pipanuran chromosome H

Similarly, further examination of Nürnberger Fig. 6 (reproduced below) suggests that:

- BVA10 \equiv XTR4q \equiv ancestral pipanuran chromosome E
 - BVA7 + part of BVA1 \equiv XTR4p \equiv ancestral pipanuran chromosome D
 - BVA12 \equiv XTR8p \equiv ancestral pipanuran chromosome J
 - Part of BVA1 \equiv XTR8q \equiv ancestral pipanuran chromosome K
 - BVA11 + part of BVA1 \equiv XTR9 \equiv ancestral pipanuran L
 - Part of BVA1 \equiv XTR10 \equiv ancestral pipanuran M
- BVA1-BVA11-BVA12 are a mix of XTR8-XTR9-XTR10

NOTE: This observation is consistent with our finding that ancestral pipanuran J, L, and M (represented by XTR 8, 9, and 10) are prone to fusion in pipanurans.

Synteny between *B. variegata* and *X. tropicalis*. Circos (v0.69-6) (Krzyszewski *et al.* 2009) plot of 737 *B. variegata* target sequences from the 12 LGs (Bv, unit is cM) aligned against the *X. tropicalis* genome assembly (Xt, unit is Mb) with BLAST+ (v. 2.9.0) (Camacho *et al.* 2009).

The patterns of fusion and breakage are very well described, but given the limited change

observed (and likely observable) it is harder to derive "significance" from the resulting patterns, and thus to define meaningful "process". I wonder what a set of frog karyotypes might look like if evolved under an entirely neutral/stochastic model of simple low rates of fusion and fission (random across chromosomes).

Thanks for this comment and suggestion! There are two ways to ask this question:

1. *Are translocations randomly distributed across phylogeny?* As shown visually in **Fig. 1** and cataloged in **Table 1**, we find a total of 17 translocations in the phylogeny of our eight genomes (counting both *X. laevis* L and S subgenomes), which spans ~1.05 billion years of total branch length. This is approximately one translocation (not distinguishing by type) every 62 million years.

From this rate, we can ask if any of the individual branches show an unusually high number of translocations relative to the overall rate. For example, on the ~200 my *L. ailaonicum* branch we would expect $200 / 62 = 3.2$ translocations, but we observe none. Assuming a Poisson distribution this would occur with probability $e^{-3.2} \sim 0.04$. This is before Bonferroni correction for the various lineages to be tested. So the lack of translocation on *L. ailaonicum* does not reject the null hypothesis of a steady, low rate of change across the entire pipanuran phylogeny.

Conversely, on the ~75 my *E. coqui* lineage (circled 4 in **Fig. 1**) we would expect ~1.2 translocations under the hypothesis of a steady slow rate of change across pipanurans, but observe six. This has an uncorrected Poisson $p = 0.001$ (probability of 6 or more translocations given a Poisson distribution with a mean of 1.2), so we can reject the steady slow rate hypothesis on this branch. This is the only branch where the steady slow rate hypothesis is rejected, and is consistent with the known karyotypic variability in the *Euleutherodactylus* lineage³⁵.

2. *Are breaks distributed randomly?* As shown in **Fig. 1**, only one of the inferred translocations splits a chromosome arm (in *P. adspersus*, producing chromosomes 3 and 6 by splitting ancestral elements A and M). The remaining events are either Robertsonian (involving breaks near a centromere) or end-to-end (near a telomere). As suggested by the reviewer we can ask what this tells us about "process": does our observation that translocations are predominantly Robertsonian or end-end reject a random breakage model? We now show that such a model can be rejected with $p \sim 4 \times 10^{-4}$.

This is further elaborated in the main text (page 16) and in a new section of **Supplementary Note 4** (page 59), copied here below for your convenience:

Propensity for centric translocation under a random break model

As shown in **Fig. 1**, only one of the 14 inferred translocation breakpoints split a chromosome arm – in *P. adspersus*, chromosomes 3 and 6 arise from a reciprocal translocation that breaks ancestral elements A and M. The remaining breaks are either Robertsonian (i.e., occur close to a centromere) or end-to-end (near a telomere). We asked whether the apparent concentration of breaks near centromeres and telomeres was consistent with a random breakage model.

The average chromosome size across our genomes (counting the L and S subgenomes of *X. laevis* separately) is 170 Mb (average genome size 1.88 Gb; $n \sim 11$ chromosomes per genome). To keep our model simple we generously assume that a break would be considered “arm preserving” if it occurred in a 10 Mb window centered on a centromere, or within 5 Mb of the end of a chromosome. Thus each chromosome contributes ~ 20 Mb of arm-preserving targets and 150 Mb of target space that would result in a non-Robertsonian or end-end event. Thus under a random break model the probability that an individual break is arm-preserving is $150 / 170 = 0.88$. Thus if the 14 breaks occurred at random along the genome, we would expect $14 \times 0.88 = 12.3$ chromosome arms to be broken, yet we observe only 2 (from the one reciprocal translocation in ancestral elements A and M that result in *P. adspersus* chromosomes 3 and 6). Under a simple Poisson random break model, the probability of 2 or fewer breaks is $(1 + 12.3 + 12.3^2/2) \times e^{-12.3} = 4 \times 10^{-4}$. We, therefore, reject a simple random break model.

Thanks for suggesting these useful “back of the envelope” calculations that help turn our observations into more concrete statements about evolutionary rate and process.

One consequence of the fission and fusion process is the rate of crossovers per DNA base (cM per Mb). There are many shifts in chromosome (arm) size on the tree: do these shifts correspond to changed evolutionary rates for the loci affected (as would be predicted if recombination is one of the major drivers of mutation)? Element H is one of several that differ between the two major clades in the tree...

This would be very interesting but is beyond our scope. One important point is that while we know that the chromosome arms shift in size, we do not know the sizes of the corresponding recombinogenic regions except in *X. tropicalis*, which would require recombination data for other frogs. It is not entirely obvious how the size of these recombinogenic regions scales with genome size. Since we do not have recombination data for the other frogs (a major effort), we cannot address this here.

The Rab1 like patterning of the interchromosomal Hi-C contacts could usefully be contextualised by the work of the Dudchenko/Erez Lieberman/Rowland labs in exploring patterns across Metazoa [2]: where do frogs fit into their models (neatly, it would seem).

Thank you for this comment. As suggested, we now include Hoencamp et al.¹ ACA scores to quantify the degree of Rabl-ness in our frog datasets. Except in one species (*E. coqui*, for which Hi-C insert coverage is possibly too low), our frog genomes show positive (i.e., type-I or “Rabl-like”) ACA scores, but these scores vary in magnitude across species and developmental stages. We have added a further discussion of these points in the main text (pages 22–23).

Because the ACA scores distort the observed contact patterns by rescaling the HiC contact map to “recenter” centromeres (as described in Hoencamp), we also implemented a more direct method to quantify the enrichment of centromere-centromere contacts (a measure of centromere clustering) and the centromere-to-telomere polarity of arm-arm contacts between chromosomes relative to centromere-telomere contacts (see new **Supplementary Fig. 15**, and new **Supplementary Table 18**). A description of this method is now included in **Supplementary Note 5**.

(Please note that, throughout the main text, we have converted all uses of “Rabl” to “Rabl-like,” as we cannot show with the current data—despite evidence of centromere-centromere and telomere-telomere clustering—that the centromeres and telomeres are situated opposite-from and near-to the nucleolus, respectively, characteristic of the *sensu stricto* Rabl configuration³.)

I would have liked to see more explicit discussion of the runs of homozygosity noted in the Xt genome

Thanks for this suggestion. We have rewritten and expanded this section (pages 10–11). Runs of homozygosity are generated by inbreeding, which leads to a steady increase in identity by descent. Residual runs of heterozygosity preserve pairs of founder haplotypes. We now provide a more detailed analysis of the residual runs of heterozygosity (**Supplementary Fig. 6**), which shows that the observed residual heterozygosity of our nominally 17th-generation inbred reference individual (11.7% of the map length) is substantially more than the 3.2% expected based on standard theory in the absence of selection³⁶. We note that this difference could be due to balancing selection, but is more likely explained by one or more errors in the (decades-long) inbreeding process since 13th-generation members of the lineage have ~44% residual heterozygosity which far exceeds the 7.4% expectation from repeated full-sib mating. We also note that the four-fold reduction in residual heterozygosity from the 13th to 17th generation is consistent with theoretical expectations for full-sib mating in the absence of selection.

1 see <https://academic.oup.com/g3journal/article/11/12/jkab286/6353606?login=true> [which includes coauthors from the current study I see]

2 see <https://www.science.org/doi/10.1126/science.abe2218>

Smaller comments.

Abstract

"phylogenetically ancient group of living amphibians"

- an odd phrasing; I see what is meant; Perhaps "Extant frogs"?

Thanks! We have changed “are a phylogenetically ancient group of living amphibians” to “are a phylogenetically ancient group of anuran amphibians”, which is clearer. There is no need to specify “living” or “extant” since it is clear from the context that we are talking about the genomes of contemporary species.

Abstract and introduction; also elsewhere

"emerging model frog species"; "the túngara frog (*Engystomops pustulosus*), which is a model for vocalization, and the Zaire dwarf clawed frog (*Hymenochirus boettgeri*), which has an unusually small embryo and is a model for regulation of cell and body sizes."; "three new emerging model species"[results]

- if it is claimed these are "models" I would expect to see a body of work referenced. I think the authors mean "would be an interesting system in which one could study...". Under this definition of "model" all species are models for their unique or shared biology). I suggest remove the assertion of model status if there is no work supporting this.

Thanks for pointing out the lack of citations to the growing body of work on these emerging models. These have been added to the main text in the introduction, and provide direct support for the “emerging model” claim.

"occurred along frog phylogeny"

- during phylogeny

Thanks. We have made this change as suggested.

"shotgun cloning and/or sequences"

- and/or sequencing

Thanks. We have corrected this typo.

"gaps in the earlier genome assemblies arising from cloning biases in the Sanger sequencing process" p8

- likely arising (this is an assertion)

We have added the qualifier “likely” as suggested since we have not done any specific follow-up work to confirm this statement.

Figure 1

It would be informative to have (perhaps as a panel in Fig1) a tree of anurans illustrating the taxonomic relationships of the various groups mentioned in the results section (Pipoidea,

Neobatrachia, Leiopelmatoidea, etc etc). This tree could also include the relative inferred timing of the splits discussed in the text.

Thanks. We would prefer not to further complicate **Fig. 1**, but have instead added to the legend the names of the relevant clades to aid the reader, and have attempted to simplify the use of Anuran phylogenetic nomenclature in the main text.

The major clades sampled in our genome sequencing effort are:

- Pipoidae: *H. boettgeri*, *Xenopus* (both family Pipidae)
- Pelobatoidea: *L. ailaonicum* (family Megaphrynidae)
Neobatrachia (Hyloidea): *E. pustulosus*, *E. coqui* (family Leptodactylidae and Euleutherodactylidae, respectively)
- Neobatrachia (Ranoidea): *P. adspersus* (family Pyxicephalidae)

The current phylogeny of these groups is (Pipoidae, (Pelobatoidea, (Neobatrachia))) as shown in **Fig. 1**. Regarding the inferred timing of the splits between these taxa, they are shown in **Fig. 1a**, inferred as described in the **Methods** and consistent with Feng et al.³⁷.

Karyotypic stability... perhaps needs more discussion - is this a taxonomically restricted trait?

Pipoidae

[https://goat.genomehubs.org/search?query=tax_tree\(30319\) AND haploid_number AND tax_rank\(species\)&result=taxon&fields=haploid_number&includeEstimates=false&summaryValues=count&taxonomy=ncbi&report=histogram&rank=species#tax_tree\(30319\) AND haploid_number AND tax_rank\(species\)](https://goat.genomehubs.org/search?query=tax_tree(30319) AND haploid_number AND tax_rank(species)&result=taxon&fields=haploid_number&includeEstimates=false&summaryValues=count&taxonomy=ncbi&report=histogram&rank=species#tax_tree(30319) AND haploid_number AND tax_rank(species))

Neobatrachia

[https://goat.genomehubs.org/search?query=tax_tree\(Neobatrachia\) AND haploid_number AND tax_rank\(species\)&result=taxon&fields=haploid_number&includeEstimates=false&summaryValues=count&taxonomy=ncbi&report=histogram&rank=species#tax_tree\(Neobatrachia\) AND haploid_number AND tax_rank\(species\)](https://goat.genomehubs.org/search?query=tax_tree(Neobatrachia) AND haploid_number AND tax_rank(species)&result=taxon&fields=haploid_number&includeEstimates=false&summaryValues=count&taxonomy=ncbi&report=histogram&rank=species#tax_tree(Neobatrachia) AND haploid_number AND tax_rank(species))

Thanks for this comment. Originally, we cited Morescalchi (1980) “Evolution and karyology of the amphibians.” *Boll. Zool.* **47**, 113–126 for the statement that variation in chromosome number among frogs is very limited. We have now also cited two more recent catalogs of this information, including the very useful new resource cited by the reviewer:

Gregory, T.R. Animal Genome Size Database. (2001). <http://www.genomesize.com>.

Sotero-Caio, C.G., Challis, R., Kumar, S. & Blaxter, M. (2021) Genomes on a Tree (GoaT): A centralized resource for eukaryotic genome sequencing initiatives. *BISS* **5**: e74138. <https://doi.org/10.3897/biss.5.74138>

We caution, however, that simple equality of chromosome numbers does not necessarily mean 1:1 correspondence between chromosomes, as shown in our study. So simple

karyotype data (in the sense of chromosome number) allows only limited conclusions to be made.

"We note that the D centromere persists in both end-to-end fusions involving D, suggesting that centromeres derived from different ancestral elements may be differentially susceptible to silencing."

- n=2 and thus this "observation" is not really worth mentioning...

This is a good point! Nevertheless, we think this observation is worth pointing out as a curiosity, with the disclaimer (page 14): "... although with only two examples this could have occurred by chance."

Figure 2

"Chromosomes are centered by the position of centromeric tandem repeats (black dot and dotted vertical line)."

- I dont see the dot and line

"the X. tropicalis centromeric satellite repeat from tandem repeat analysis with a red star"

- not visible in chr7?

Thanks, these have been corrected:

- For **Fig. 2** the opacity of the black dot has been increased to 100% so that it is now clearly visible, and for simplicity the vertical line removed.
- For **Fig. 3a**, the two stars overlap on chromosome 7, and this is now noted in the figure legend.

Heterozygosity

"15 long heterozygous blocks ranging in size from 1.34 to 74.6 Mb."

- the relevance of the sequencing of geographic populations is not made clear in the main text; are these retained Hz blocks derived specifically from one population or the other? Are they evident in the parental populations? How "related" are the presumed parental populations to the individuals set up to start the inbreeding? Is this more or less than one would expect from the inbreeding (this is modellable)?

These are two separate points.

First, the long heterozygous blocks in the reference genotype shown in **Supplementary Fig. 6** are those that remain after 17 generations of brother-sister mating, starting from a pair of Nigerian founders. These residual blocks of heterozygosity are more extensive than would be expected under repeated full-sib mating. This quantitative discrepancy suggests either (1) selective retention of heterozygosity due to balancing selection during inbreeding, or (more mundanely) mistakes during the decades-long process. The latter is consistent with high levels of heterozygosity observed in 13th-generation members of the lineage; the four-

fold reduction from the 13th generation to our 17th-generation reference is consistent with expectation in the absence of selection. We have expanded this discussion to make these points clear (pages 10–11).

Whatever the reasons for their retention, however, these residual blocks of heterozygosity are due to haplotypes transmitted from the wild Nigerian founders, so we can use these blocks to estimate heterozygosity in the Nigerian population (i.e., between the wild-caught parents of this inbred line). This is now more clearly stated in the main text (page 11).

Second, independent of the reference genotype, we also report shotgun sequencing from three pools of animals: 12 males and 8 females from a colony of Nigerian frogs maintained for 13 generations), and 6 F7 Ivory Coast “B” males. These are now discussed in **Supplementary Notes 1 and 2**. These two populations are the main source of experimental *X. tropicalis* animals. The depth of the Ivory Coast pool was very low (mean depth four, as now noted in **Supplementary Note 1**), and so we only quote the total number of variant sites detected rather than a *bona fide* heterozygosity rate per kilobase. This data is provided as a resource for the research community.

References

1. Hoencamp, C. *et al.* 3D genomics across the tree of life reveals condensin II as a determinant of architecture type. *Science* **372**, 984–989 (2021).
2. Rabl, C. Über Zelltheilung. *Morphologisches Jahrbuch* 214–330 (1885).
3. Muller, H., Gil, J., Jr & Drinnenberg, I. A. The impact of centromeres on spatial genome architecture. *Trends Genet.* **35**, 565–578 (2019).
4. Varoquaux, N. *et al.* Accurate identification of centromere locations in yeast genomes using Hi-C. *Nucleic Acids Res.* **43**, 5331–5339 (2015).
5. Bunnik, E. M. *et al.* Comparative 3D genome organization in apicomplexan parasites. *Proc. Natl. Acad. Sci. U. S. A.* **116**, 3183–3192 (2019).
6. Mudd, A. B., Bredeson, J. V., Baum, R., Hockemeyer, D. & Rokhsar, D. S. Muntjac chromosome evolution and architecture. *Cold Spring Harbor Laboratory* 772343 (2019) doi:10.1101/772343.
7. Rao, S. S. P. *et al.* A 3D map of the human genome at kilobase resolution reveals principles of chromatin looping. *Cell* **159**, 1665–1680 (2014).
8. Darrow, E. M. *et al.* Deletion of DXZ4 on the human inactive X chromosome alters higher-order genome architecture. *Proc. Natl. Acad. Sci. U. S. A.* **113**, E4504–12 (2016).
9. Vian, L. *et al.* The Energetics and Physiological Impact of Cohesin Extrusion. *Cell* **175**, 292–294 (2018).
10. Peart, C. R. *et al.* Hi-C scaffolded short- and long-read genome assemblies of the California sea lion are broadly consistent for syntenic inference across 45 million years of evolution. *Mol. Ecol. Resour.* **21**, 2455–2470 (2021).
11. DNA Zoo Assemblies. *DNA Zoo* <https://www.dnazoo.org/assemblies> (2018).
12. Dobrynin, P. *et al.* Genomic legacy of the African cheetah, *Acinonyx jubatus*. *Genome Biol.* **16**, 277 (2015).
13. Hu, Y. *et al.* Comparative genomics reveals convergent evolution between the bamboo-eating giant and red pandas. *Proc. Natl. Acad. Sci. U. S. A.* **114**, 1081–1086 (2017).
14. Zhang, C. *et al.* tagHi-C Reveals 3D Chromatin Architecture Dynamics during Mouse Hematopoiesis. *Cell Rep.* **32**, 108206 (2020).
15. Stevens, T. J. *et al.* 3D structures of individual mammalian genomes studied by single-cell Hi-C. *Nature* **544**, 59–64 (2017).
16. O'Connor, R. E. *et al.* Reconstruction of the diapsid ancestral genome permits chromosome evolution tracing in avian and non-avian dinosaurs. *Nat. Commun.* **9**, 1883 (2018).
17. Bush, G. L., Case, S. M., Wilson, A. C. & Patton, J. L. Rapid speciation and chromosomal evolution in mammals. *Proc. Natl. Acad. Sci. U. S. A.* **74**, 3942–3946 (1977).
18. Ferguson-Smith, M. A. & Trifonov, V. Mammalian karyotype evolution. *Nat. Rev. Genet.* **8**, 950–962 (2007).
19. Damas, J. *et al.* Evolution of the ancestral mammalian karyotype and syntenic regions. *Proc. Natl. Acad. Sci. U. S. A.* **119**, e2209139119 (2022).
20. Wright, S. On the Probability of Fixation of Reciprocal Translocations. *Am. Nat.* **75**, 513–522 (1941).
21. Lande, R. The fixation of chromosomal rearrangements in a subdivided population with

- local extinction and colonization. *Heredity* **54 (Pt 3)**, 323–332 (1985).
22. Nakatani, Y., Takeda, H., Kohara, Y. & Morishita, S. Reconstruction of the vertebrate ancestral genome reveals dynamic genome reorganization in early vertebrates. *Genome Res.* **17**, 1254–1265 (2007).
 23. Simakov, O. *et al.* Deeply conserved synteny resolves early events in vertebrate evolution. *Nat Ecol Evol* **4**, 820–830 (2020).
 24. Waters, P. D. *et al.* Microchromosomes are building blocks of bird, reptile, and mammal chromosomes. *Proc. Natl. Acad. Sci. U. S. A.* **118**, (2021).
 25. Wells, D. E. *et al.* A genetic map of *Xenopus tropicalis*. *Dev. Biol.* **354**, 1–8 (2011).
 26. Mácha, J. *et al.* Deep ancestry of mammalian X chromosome revealed by comparison with the basal tetrapod *Xenopus tropicalis*. *BMC Genomics* **13**, 315 (2012).
 27. Mitros, T. *et al.* A chromosome-scale genome assembly and dense genetic map for *Xenopus tropicalis*. *Dev. Biol.* **452**, 8–20 (2019).
 28. Dernburg, A. F. *et al.* Perturbation of nuclear architecture by long-distance chromosome interactions. *Cell* **85**, 745–759 (1996).
 29. Hiraoka, Y. *et al.* The onset of homologous chromosome pairing during *Drosophila melanogaster* embryogenesis. *J. Cell Biol.* **120**, 591–600 (1993).
 30. Marshall, W. F., Dernburg, A. F., Harmon, B., Agard, D. A. & Sedat, J. W. Specific interactions of chromatin with the nuclear envelope: positional determination within the nucleus in *Drosophila melanogaster*. *Mol. Biol. Cell* **7**, 825–842 (1996).
 31. Payne, A. C. *et al.* In situ genome sequencing resolves DNA sequence and structure in intact biological samples. *Science* **371**, (2021).
 32. Aguirre-Lavin, T. *et al.* 3D-FISH analysis of embryonic nuclei in mouse highlights several abrupt changes of nuclear organization during preimplantation development. *BMC Dev. Biol.* **12**, 30 (2012).
 33. Hou, C., Li, L., Qin, Z. S. & Corces, V. G. Gene density, transcription, and insulators contribute to the partition of the *Drosophila* genome into physical domains. *Mol. Cell* **48**, 471–484 (2012).
 34. Nürnberger, B. *et al.* A dense linkage map for a large repetitive genome: discovery of the sex-determining region in hybridizing fire-bellied toads (*Bombina bombina* and *Bombina variegata*). *G3* **11**, (2021).
 35. Bogart, J. P. & Hedges, S. B. Rapid chromosome evolution in Jamaican frogs of the genus *Eleutherodactylus* (Leptodactylidae). *J. Zool.* **235**, 9–31 (1995).
 36. Nagylaki, T. *Introduction to Theoretical Population Genetics*. (Springer Berlin Heidelberg, 1992).
 37. Feng, Y.-J. *et al.* Phylogenomics reveals rapid, simultaneous diversification of three major clades of Gondwanan frogs at the Cretaceous-Paleogene boundary. *Proc. Natl. Acad. Sci. U. S. A.* **114**, E5864–E5870 (2017).

REVIEWERS' COMMENTS

Reviewer #1 (Remarks to the Author):

In the revised manuscript (Bredeson, Mudd, Medina-Ruiz et al., "Conserved chromatin and repetitive patterns reveal slow genome evolution in frogs"), the authors have clarified a number of points and incorporated comparative findings that provide essential context for interpretation. My original questions and concerns have largely been addressed, and I recommend publication without additional revision.

The authors have deftly integrated genomic, genetic, and chromatin-based datasets across multiple lineages, and the manuscript offers a model for future analyses. These results will be immediately useful for contemporary studies, and this paper as a whole will have enduring value for our understanding of vertebrate genome organization and evolution.

Reviewer #2 (Remarks to the Author):

The authors have ably and effectively addressed all my queries and musings on their excellent manuscript. I am happy to recommend publication.

It was a particular pleasure to read a rebuttal document that was neither defensive nor dismissive.

I found only a single remaining "grammatical infelicity":

423ff "However, the centromeres of acrocentric chromosomes lie within 30 Mb of the telomeres, which precludes the extended subtelomere-associated repeats (Fig. 2 and Supplementary Fig. 11)."

... precludes may be the wrong word, or needs to be conditioned by stating what the s-t repeats are precluded from.

REVIEWERS' COMMENTS

Reviewer #1 (Remarks to the Author):

In the revised manuscript (Bredeson, Mudd, Medina-Ruiz et al., “Conserved chromatin and repetitive patterns reveal slow genome evolution in frogs”), the authors have clarified a number of points and incorporated comparative findings that provide essential context for interpretation. My original questions and concerns have largely been addressed, and I recommend publication without additional revision.

The authors have deftly integrated genomic, genetic, and chromatin-based datasets across multiple lineages, and the manuscript offers a model for future analyses. These results will be immediately useful for contemporary studies, and this paper as a whole will have enduring value for our understanding of vertebrate genome organization and evolution.

Thanks!

Reviewer #2 (Remarks to the Author):

The authors have ably and effectively addressed all my queries and musings on their excellent manuscript. I am happy to recommend publication.

It was a particular pleasure to read a rebuttal document that was neither defensive nor dismissive.

Thanks!

I found only a single remaining "grammatical infelicity":

423ff "However, the centromeres of acrocentric chromosomes lie within 30 Mb of the telomeres, which precludes the extended subtelomere-associated repeats (Fig. 2 and Supplementary Fig. 11)."

... precludes may be the wrong word, or needs to be conditioned by stating what the s-t repeats are precluded from.

Thank you for calling our attention to this sentence. We have corrected the grammatical error to improve its clarity (revised main text, pg. 19, lines 420–422):

“However, the centromeres of acrocentric chromosomes lie within 30 Mb of telomeres and preclude the presence of extended subtelomere-associated repeats (**Fig. 2 and Supplementary Fig. 11**).”